# Biosynthesis and bioactivity of anti-inflammatory triterpenoids in *Calendula officinalis*

D. Golubova [1,2,4], M. Salmon [1,4] ✉, H. Su[1,3,4], C. Tansley [1,3], GG Kaithakottil [1], G. Linsmith[1], C. Schudoma [1], D. Swarbreck [1], MA O'Connell[2] ✉ & NJ Patron [1,3] ✉

Plants have been central to traditional medicine for millennia, yet the precise metabolites responsible for their therapeutic properties often remain unidentified. In this work, we investigate the reported anti-inflammatory properties of *Calendula officinalis* (pot marigold), an ancient medicinal herb. We confirm C16-hydroxylated triterpenoids as key contributors to the anti-inflammatory activity of *C. officinalis* floral extracts and uncover a mechanism by which they act in modulating interleukin 6 release. Through biosynthetic pathway elucidation, we demonstrate that the oxidosqualene synthase catalysing the first committed step emerged early in Asteraceae evolution and identify residues governing product specificity. Further, we functionally characterise cytochrome P450s and acyltransferases responsible for downstream modifications. By reconstructing the complete biosynthetic pathway in the plant chassis *Nicotiana benthamiana*, we provide a basis for the future bioproduction of the anti-inflammatory components. Our work highlights how integrated studies of bioactivity and biosynthesis can unlock the therapeutic potential of medicinal plants.

The Asteraceae (aster) family is one of the largest families of flowering plants with an estimated 23,000–35,000 species[1]. Their history of use in medicine dates back thousands of years; written references to the medicinal uses of *Calendula officinalis* (pot marigold) were recorded by Pliny the Elder[2]. Indeed, species in the *Calendula* genus remained in medical use until the twentieth century. For example, until 1942, pot marigold was listed in the British Pharmacopoeia and the United States National Formulary, which provide official standards for pharmaceutical substances[3].

In the modern era, extracts of numerous Asteraceae species have been demonstrated to exhibit antibacterial, antiviral, antifungal, anti-inflammatory, and antiplasmodial bioactivities[4]. However, in only a few species has bioactivity been unequivocally associated with a specific compound. Of these, the anti-plasmodial bioactivity of artemisinin,

found in the glandular trichomes of *Artemisia annua* L. (sweet wormwood) is perhaps the most famous[5]. Nonetheless, drawing on their history of use in traditional medicine, extracts of many Asteraceae species are commercially exploited: extracts of pot marigold are used in a wide range of skin care products.

Pot marigold extracts have been reported to contain numerous compounds, most notably triterpenoids. These include oleanolic acid saponins (glucosides or glucuronides), α-amyrin, β-amyrin, lupeol, and triterpene fatty acid esters (TFAEs), of which faradiol myristate and faradiol palmitate are the most abundant[6] (Fig. 1). Triterpenoids are one of the largest classes of secondary metabolites in plants, with more than 14,000 types of structures described[7], many of which are bioactive and several of which are in use as vaccine adjuvants, anti-cancer drugs and food sweeteners[8]. Anti-inflammatory bioactivity has been

[1]Engineering Biology, Earlham Institute, Norwich Research Park, Norwich, UK. [2]School of Chemistry, Pharmacy and Pharmacology, University of East Anglia, Norwich, UK. [3]Department of Plant Sciences, University of Cambridge, Downing Street, Cambridge, UK. [4]These authors contributed equally: D. Golubova, M. Salmon, H. Su. ✉e-mail: melissa.salmon@earlham.ac.uk; m.oconnell@uea.ac.uk; njp56@cam.ac.uk

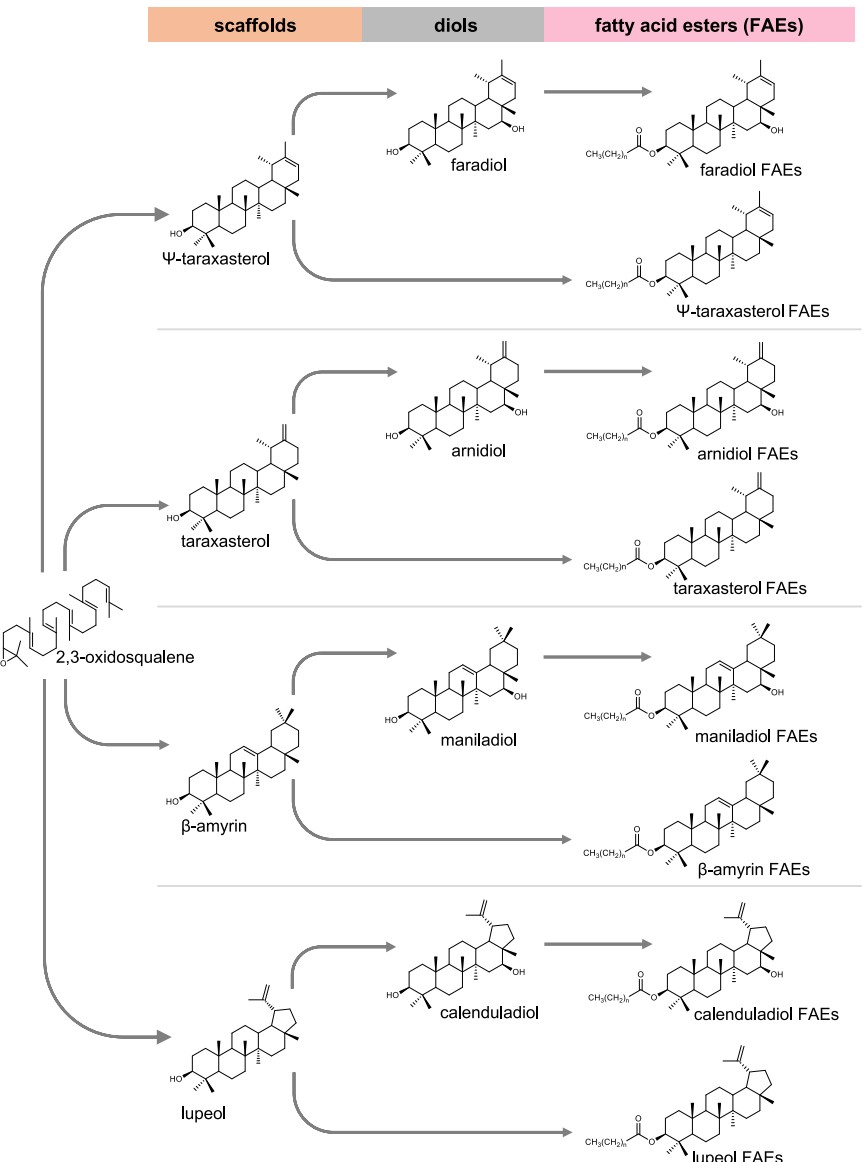

**Fig. 1 | Pentacyclic triterpene scaffolds, diols and fatty acid esters found in extracts of *Calendula officinalis*.** Fatty acid ester groups include laurate, myristate and palmitate.

reported for several pentacyclic triterpenes, particularly lupeol[9] and betulin[10], which are ingredients in an FDA-approved wound-healing hydrogel, FILSUVEZ®.

Early studies observed that extracts of pot marigold, as well triterpenoids present in those extracts, could reduce oedema in mouse ear inflammation models[11,12]. Further studies showed that the triterpene diols faradiol and arnidiol had more anti-oedematous activity than the less-polar Ψ-taraxasterol, tarax-asterol and their FAEs[13]. Pot marigold floral extracts also reduced inflammation in acute and chronic mouse models of paw oedema and sepsis[14]. These anti-inflammatory effects may be due to a variety of mechanisms. For example, pot marigold flower extracts stimulate fibroblast proliferation and migration[15]. In addition, they inhibit production of the pro-inflammatory cytokines interleukin-1 beta (IL-1β), interleukin 6 (IL-6) and tumour necrosis factor-alpha (TNF-α) in response to lipopolysaccharide (LPS) in macrophages in vitro and in vivo[16]. Furthermore, in gastric epithelial cells, fractions rich in triterpene diols and TFAEs inhibited the activity of the transcription factor NF-κB, which is a pivotal mediator of inflam-matory responses and regulator of these pro-inflammatory

cytokines[17]. In contrast, more recently, it was shown that floral extracts of pot marigold increased NF-κB DNA-binding and the pro-inflammatory chemokine interleukin 8 in human keratinocytes[18]. This study also reported that the pot marigold triterpenes were unable to modulate NF-κB DNA-binding, leading the authors to conclude that triterpenes play a minor role in the anti-inflammatory bioactivity of pot marigold extracts and that fur-ther studies are necessary to evaluate which constituents are responsible for this activity.

Most triterpene scaffolds are biosynthesised from the linear 2,3-oxidosqualene by oxidosqualene cyclases (OSCs) (Fig. 1). These scaf-folds are then decorated by various enzymes to produce the enormous diversity of triterpenoids[19]. The most abundant triterpenoids found in pot marigold are derived from the pentacyclic triterpene scaffold, Ψ-taraxasterol (Fig. 1). To date, OSCs that produce Ψ-taraxasterol as the primary product have yet to be characterised. Enzymes identified from *Taraxacum kok-saghyz*, *Taraxacum coreanum* (Russian and Korean dandelion, respectively)[20,21] and *Lactuca sativa* (lettuce)[22] pre-dominantly produce a related scaffold, taraxasterol, as well as limited quantities of Ψ-taraxasterol.

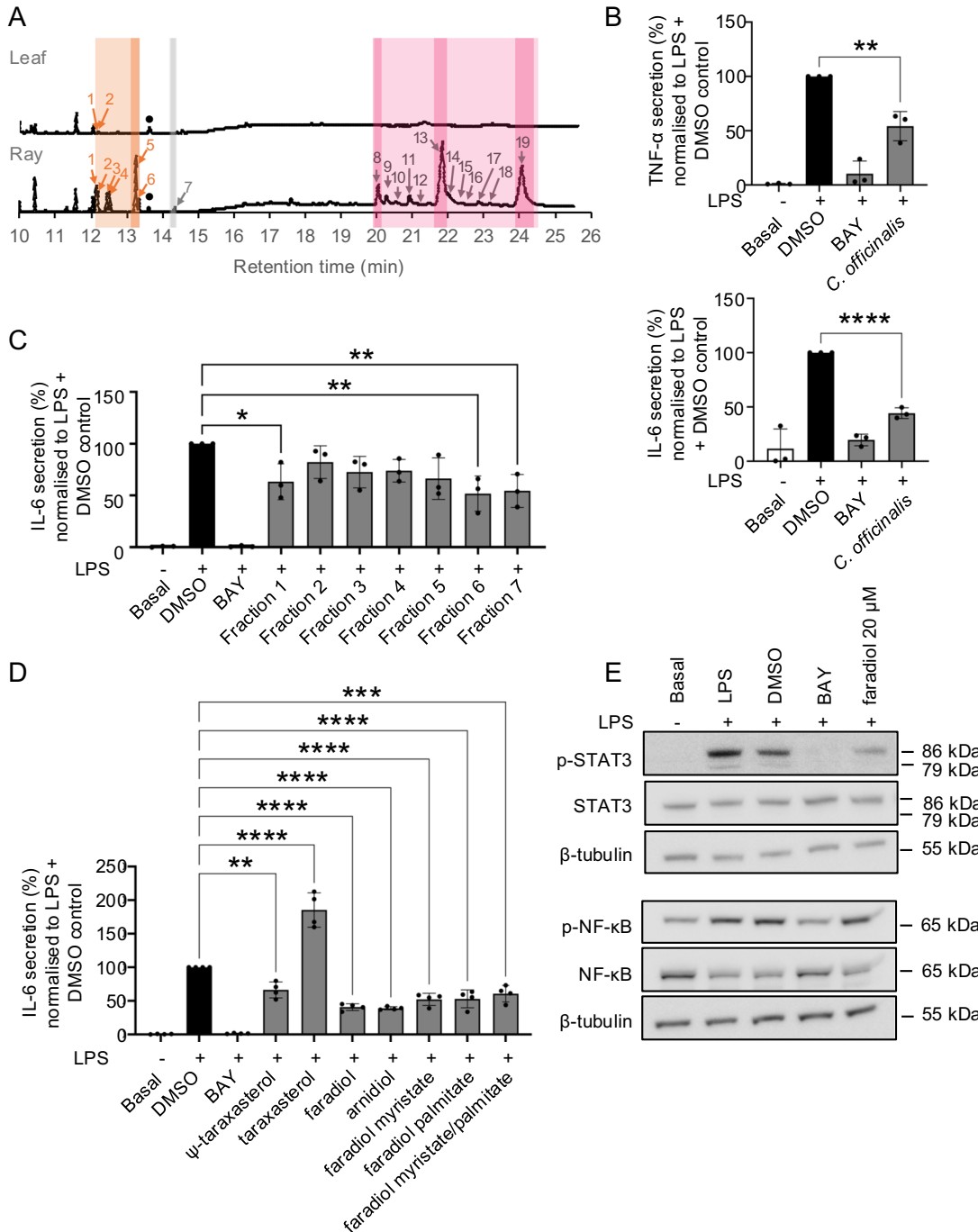

**Fig. 2 | The anti-inflammatory activity of *Calendula officinalis* (pot marigold) floral extracts and triterpenoids. A** Total ion chromatogram showing spectral peaks identified by GC-MS analysis of pot marigold leaf and ray floret tissues. Metabolite groups are highlighted as follows: triterpene monols (orange); triterpene diols (grey); triterpene fatty acid esters (magenta). Faradiol esters and their precursors are highlighted in a darker shade. Metabolite amounts were determined by peak area analysis using friedelin as an internal standard (•); 1, β-amyrin; 2, isofucosterol; 3, α-amyrin; 4, lupeol; 5, ψ-taraxasterol; 6, taraxasterol; 7, faradiol; 8, faradiol/arnidiol laurate; 9, β-amyrin myristate; 10, maniladiol myristate; 11, calenduladiol myristate; 12, ψ-taraxasterol/taraxasterol myristate; 13, faradiol/arnidiol myristate; 14, β-amyrin palmitate; 15, lupeol palmitate; 16, maniladiol palmitate; 17, calenduladiol palmitate; 18, ψ-taraxasterol/taraxasterol palmitate; 19 faradiol/ arnidiol palmitate. **B** The anti-inflammatory effect of pot marigold floral extracts (50 μg/mL) on the release of TNF-α (above) and IL-6 (below) in LPS-induced human

monocytic THP-1 cells. Error bars illustrate the mean and standard error of three biological replicates. **C** Anti-inflammatory effects of seven fractions of pot marigold extracts on the release of IL-6 in LPS-activated THP-1 cells. Details of fractionation and analysis of fractions are provided in Supplementary Figs. 6–8. Error bars illustrate the mean and standard error of 3 biological replicates. **D** Anti-inflammatory effect of different triterpenoids (20 μM) on the release of IL-6 in LPS-activated THP-1 cells. Error bars illustrate the mean and standard error of 4 biological replicates. **E** Effect of faradiol (20 μM) on NF-κB and STAT3 signalling pathways in LPS-induced human monocytic THP-1 cells. BAY: BAY 11-7082, (E)-3-(4-Methyl-phenylsulfonyl)-2-propenenitrile (positive control); DMSO (solvent control). Statistical significance was determined using one-way ANOVA with a Post-hoc Dunnett test to LPS + DMSO; *$p < 0.0332$, **$p < 0.0021$, ***$p < 0.0002$, ****$p < 0.0001$. $P$ values are provided in Supplementary Data 3.

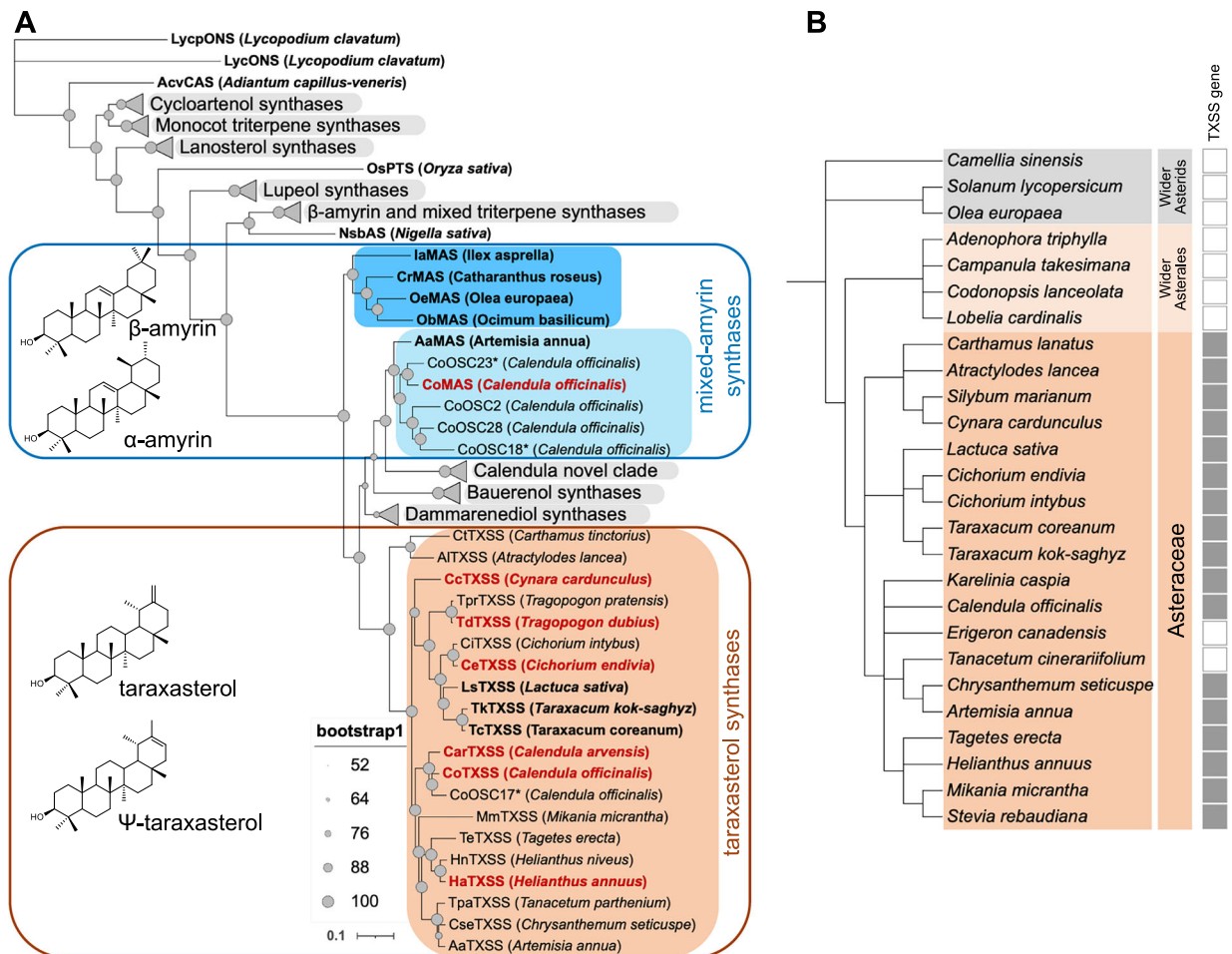

**Fig. 3 | Phylogenetic analysis of plant oxidosqualene cyclases (OSCs). A** Maximum-likelihood tree of characterised plant OSCs constructed in IQTree using the JTT matrix-based model. Taraxasterol synthases (TXSSs) are highlighted in orange and mixed-amyrin synthases are highlighted in light blue (Asteraceae) and blue (non-Asteraceae) and annotated with chemical structures of their major products. A full tree showing collapsed clades (grey triangles) is provided in Supplementary Fig. 17 and a list of taxa and accession numbers are provided in Supplementary Data 6. Functionally characterised OSCs are shown in bold, those in red are characterised in this paper (Supplementary Fig. 20). An asterisk (*) indicates likely pseudogenes. Filled grey circles indicate bootstrap supports for each node. The scale bar represents the number of substitutions per site. **B** Species cladogram of Asteraceae and related species with fully sequenced genomes. Filled box indicates presence of a *TXSS* gene in the genome or transcriptome, unfilled box indicates absence in the genome. ONS = onocerin synthase; CAS = cycloartenol synthase; PTS = poacetapetol synthase; bAS=β-amyrin synthase; MAS = mixed amyrin synthase; TXSS = taraxasterol/Ψ-taraxasterol synthase.

In this manuscript, we show that faradiol fatty acid esters are a major contributor to the anti-inflammatory bioactivity of pot marigold floral extracts and provide evidence of a mechanism of action for the non-acylated diol, faradiol. Having confirmed the bioactivity of these compounds, we elucidate the genetic basis of their biosynthesis in pot marigold, characterising the function and evolution of each enzyme and reconstructing the pathway in the model plant, *Nicotiana benthamiana*.

## Results

### The anti-inflammatory activity of C:16 hydroxylated triterpenoids

To quantify the composition and distribution of triterpenes in extracts of pot marigold we conducted metabolic profiling of extracts of leaf, root, disc and ray florets using gas chromatography/mass spectrometry (GC-MS) (Fig. 2A; Supplementary Fig. 1; Supplementary Table 1) and liquid-chromatography/mass spectrometry (LC-MS) (Supplementary Fig. 2). Triterpene monols and diols were identified by comparing retention times and mass fragmentation patterns with those of authentic standards and quantified using an internal reference (Supplementary Figs. 3–5). Faradiol palmitate was purified by fractionation

(Supplementary Figs. 6–8) and the structure was confirmed by NMR (Supplementary Figs. 9–14; Supplementary Data 1 and 2). Triterpene fatty acid esters were exclusively found in floral tissue, with ray florets containing approximately 8-fold higher amounts of TFAEs than disc florets. In ray florets, faradiol myristate and faradiol palmitate were the most abundant TFAEs (13.18 and 12.46 mg/g dry weight, respectively). Faradiol esters constituted the majority (77%) of TFAEs observed in ray florets. No TFAEs were detected in leaf or root tissue (Fig. 2A; Supplementary Fig. 1; Supplementary Table 1). In addition, LC-MS confirmed the previously reported presence of oleanane-glucosides and glucuronides (Supplementary Fig. 2).

To investigate the anti-inflammatory activity, the ability of extracts to inhibit the release of the pro-inflammatory cytokines, TNF-α and IL-6, in lipopolysaccharide (LPS)-activated human monocytic (THP-1) cells was assessed by enzyme-linked immunosorbent assays (ELISAs). BAY 11-7082, (E)-3-(4-methylphenylsulfonyl)-2-propenenitrile was used as a positive control. Ethyl acetate extracts of floral tissues had negligible effects on cell viability compared to the solvent control dimethyl sulfoxide (DMSO) (Supplementary Fig. 15). Extracts showed a concentration-dependent repression of pro-inflammatory cytokines; 50 μg/mL extracts reduced LPS-activated TNF-α and IL-6 release by

46% and 56%, respectively (Fig. 2B; Supplementary Fig. 16; Supplementary Data 3).

To investigate the contribution of triterpenes to the bioactivity of floral extracts, the extracts were fractionated using liquid chromatography (LC) to obtain seven fractions. The major compounds detected were fatty acids (fractions 1, 2, 3 and 7), triterpene monols (fraction 4), and faradiol fatty acid esters (fractions 5 and 6) (Supplementary Fig. 8). Fraction 6 (containing faradiol-FAEs) as well as fractions 1 and 7 (containing fatty acids) showed significant anti-inflammatory activity, inhibiting the release of LPS-induced IL-6 (Fig. 2C; Supplementary Data 3). To investigate the activity of specific pot marigold triterpenes, we compared the activity of taraxasterol, Ψ-taraxasterol, arnidiol, faradiol, faradiol myristate, and faradiol palmitate. Unexpectedly, taraxasterol showed pro-inflammatory activity, enhancing LPS-induced IL-6 release. All other tested compounds (Ψ-taraxasterol, faradiol, arnidiol, faradiol myristate, faradiol palmitate and mixture of esters) displayed significant anti-inflammatory activity reducing LPS-induced IL6 release. The C:16 hydroxylated compounds (faradiol and arnidiol) demonstrated the strongest anti-inflammatory activity, and significantly inhibited LPS-induced IL6 release by 59% and 61%, respectively. No synergistic effect was noted with a combination of faradiol myristate and faradiol palmitate, compared to the activity of individual compounds (Fig. 2D; Supplementary TaData 3). For faradiol, concentration-dependent activity was observed between 5 µM and 20 µM (Supplementary Fig. 16).

To identify the mechanism by which faradiol regulates IL-6 release, we investigated the effect of faradiol on the phosphorylation of two transcription factors, NF-κB and STAT3, by Western blotting with primary antibodies to the phosphorylated and unphosphorylated proteins. Treatment of THP-1 cells with 20 µM of faradiol reduced phosphorylation of STAT3 but not NF-κB p65 (Fig. 2E).

Having confirmed the anti-inflammatory bioactivity of faradiol and its esters, we next sought to uncover the genetic basis of their production in pot marigold, with the aim of enabling pathway reconstruction in heterologous hosts.

## Identification of candidate Ψ-taraxasterol synthases
Faradiol is derived from a Ψ-taraxasterol triterpene scaffold, differing from taraxasterol only in the structure of the final E ring (Fig. 1). Enzymes responsible for synthesizing taraxasterol have been characterised in three species of the Cichorioideae subfamily: Russian and Korean dandelions and lettuce.

To identify a candidate taraxasterol/Ψ-taraxasterol synthase (TXSS), we first sequenced the pot marigold nuclear genome using PacBio HiFi reads polished with 10X linked-reads and scaffolded with Omni-C data. A detailed description of sequencing and genome assembly is provided in the Supplementary Methods. This produced an assembly of 1.3 Gb, consisting of 2,811 contigs with an N50 of 80.2 Mb and including 16 highly contiguous scaffolds over 25 Mb in length (Supplementary Table 2). Previous cytological studies have shown that *C. officinalis* has an allotetraploid background[23]. A k-mer analysis showed three clear peaks consistent with an allotetraploid background. We observed low heterozygosity within haplotype pairs but significant variation between the pairs which supports an ancient hybridisation event. BUSCO (Benchmarking Universal Single-Copy Orthologs) analysis indicated most genes are duplicated with a proportion that are single-copy, indicating the genome is likely to have undergone some diploidisation (Supplementary Table 2). We also sequenced and assembled transcriptomes of leaf, disc floret and ray floret tissues harvested from pot marigold and *Calendula arvensis* (field marigold).

*T. kok-saghyz* taraxasterol synthase (TkTXSS) and *Artemisia annua* (sweet wormwood) cycloartenol synthase (AaCAS) were used as queries to search the genome and transcriptome of pot marigold, identifying 27 candidate genes encoding OSCs *(CoOSCs)*. Transcripts

from 16 of these genes were identified in the transcriptome data obtained from leaves and floral tissues (Supplementary Table 3; Supplementary Data 4 and 5). We next performed a phylogenetic analysis of the translated proteins of all 27 retrieved gene sequences together with 133 previously characterised plant OSCs (Fig. 3A; Supplementary Fig. 17; Supplementary Data 6).

Almost all CoOSCs were observed to be most closely related to another CoOSC, with each pair having between 77% and 92% sequence identity. Nine of these gene pairs were located in regions with conserved synteny suggesting homology, while two gene pairs and one triplet were located adjacent to each other on the same contig (Supplementary Fig. 18; Supplementary Data 4). Consistent with diploidisation suggested by the BUSCO analysis, one of each gene pair was not expressed and comparative analysis of the intron-exon structure, sequence and conserved catalytic motifs indicated these non-expressed *OSCs* were likely to be non-functional pseudogenes (Supplementary Data 4 and 5; Supplementary Table 3). In contrast, transcripts corresponding to all five of the *CoOSCs* genes predicted to encode cycloartenol synthases were identified among the 16 *OSCs* for which expression was detected in leaf and floral tissues (Supplementary Fig. 17; Supplementary Table 3; Supplementary Data 4). The expression of other *CoOSCs* for which transcripts were not present in our leaf and floral transcriptomes may either be limited to other tissues (e.g., roots) or in response to specific stimuli.

Seven CoOSCs were in clades containing proteins involved in plant sterol biosynthesis, with the remainder belonging to clades involved in pentacyclic triterpenoid biosynthesis (Fig. 3A; Supplementary Fig. 17; Supplementary Data 6). One strongly supported clade included previously characterised TXSSs from lettuce, Russian and Korean dandelion as well as two sequences each from pot marigold, field marigold and sequences from several other Asteraceae (candidate CoTXSSs) (Fig. 3A; Supplementary Fig. 17; Supplementary Data 6). Differential expression analysis of the corresponding *CoOSCs* genes indicated that nine were mainly expressed in leaves, three were expressed in all tissues, and four were predominantly expressed in floral tissues, including one candidate *CoTXSS*; the other was not expressed (Supplementary Fig. 19).

## TXSSs evolved from a multifunctional amyrin synthase
To investigate the evolutionary origins of TXSS we first surveyed the presence of *TXSS* in publicly available plant genomes. Candidate *TXSS* genes were only identified in the genomes of the Asteraceae, including species from all three major Asteraceae subfamilies (Carduoideae, Cichorioideae and Asteroideae). No candidates were found in other plant lineages including the non-Asteraceae Asterales (Fig. 3). To characterise candidate TXSSs, we synthesised and cloned the coding sequences of genes from pot marigold (*CoTXSS*), field marigold (*CarTXSS*), *Cynara cardunculus* (globe artichoke; *CcTXSS*), *Helianthus annuus* (common sunflower; *HaTXSS*), *Cicorium endivia* (endive; *CeTXSS*), *Tragopogon dubius* (yellow salsify; *TdTXSS*), *Lactuca sativa* (lettuce; *LsTXSS*), Russian dandelion (*TkTXSS*) and Korean dandelion (*TcTXSS*) into binary expression vectors with a constitutive promoter. These constructs were transformed into *Agrobacterium tumefaciens* and co-infiltrated into the leaves of *Nicotiana benthamiana* with strains expressing a truncated HMGR (tHMGR) and a suppressor of silencing (P19). GC-MS analyses of the extracts of infiltrated leaves indicated that the primary products of all enzymes were either taraxasterol or Ψ-taraxasterol, with the ratio of these two products differing between species, which is further investigated below. All enzymes also produced trace quantities of β-amyrin and lupeol (Supplementary Fig. 20).

Phylogenetic analysis indicated that TXSSs are most closely related to mixed amyrin synthases (MASs) from the Asteraceae, bauerenol synthases and dammarenediol synthases. Basal to these is a clade containing MASs from diverse plant taxa, reported to produce either α- or β-amyrin as the major product (Fig. 3A and Supplementary

Fig. 17). This suggests that TXSSs are descended from an MAS present in an early ancestor of the Asteraceae. We therefore sought to compare MAS and TXSS sequences to identify residues important for activity in taraxasterol/Ψ-taraxasterol production. We inferred structural models of CoMAS and CoTXSS using AlphaFold2[24], both of which showed high structural similarity (RMSD = 0.806 and = 0.778, respectively) to the crystal structure of human lanosterol synthase (PDB:1W6K)[25]. In the formation of pentacyclic triterpenes, the 2,3-oxidosqualene precursor is folded within the OSC enzyme which catalyses cyclisations that form the four-ringed dammarenyl cation. The enzyme then directs a series of ring expansion and closures that determine the structure of the fifth ring. In the production of amyrins, this results in a lupanyl cation which undergoes a ring expansion to create either an ursanyl cation (deprotonated to α-amyrin) or an oleanyl cation (deprotonated to β-amyrin). To produce taraxasterol/Ψ-taraxasterol, a methyl shift of the oleanyl cation creates the final taraxasteryl cation (Fig. 4A).

Based on the position of lanosterol in 1W6K, we predicted the active site of CoMAS and CoTXSS and manually docked the taraxasteryl cation (Fig. 4B). We then compared residues within 12 Å of the predicted active sites, identifying two sites that differed between MASs and TXSSs: I367 and E371 (Fig. 4C). These residues were mutated in CoMAS, substituting them with those found in CoTXSS. The product profiles of the single and double mutants were compared to those obtained from the wild-type enzymes following transient expression in *Nicotiana benthamiana* (Fig. 4D). CoMAS[I367M] produced significantly less α/β-amyrin and significantly more taraxasterol/Ψ-taraxasterol than CoMAS (Fig. 4E; Supplementary Data 7). CoMAS[E371D] also produced significantly more taraxasterol/ψ-taraxasterol than CoMAS though α/β-amyrin production was not significantly affected (Fig. 4E; Supplementary Data 7). CoMAS[I367M,E371D] produced significantly less α/β-amyrin than CoMAS and the single mutants and similar levels of taraxasterol/Ψ-taraxasterol as those produced by CoTXSS (Fig. 4E; Supplementary Data 7).

These data, together with the phylogenetic position of the clade, suggest that TXSSs evolved by duplication and neofunctionalisation of a MAS following the divergence of the Asteraceae. To further investigate this, we dated the origin of the TXSS clade using Bayesian phylogenetic analysis with a strict clock and two fossil calibrations. These data suggested that TXSSs emerged 83-121 mya (Supplementary Fig. 21). Supporting an origin of TXSS by duplication, *CoTXSS* is adjacent to a gene predicted to encode an OSC that is closely related to CoMAS (CoOSC2) (Fig. 3 and Supplementary Fig. 17), but for which transcripts were not detected in our leaf and floral transcriptomes. Further, the syntenic organisation of this genomic region is conserved in other Asteraceae (Supplementary Fig. 22).

### The product specificity of TXSSs varies across Asteraceae lineages

As noted above, we observed differences in the product profiles of TXSSs from different species, notably enzymes from species of Carduoideae, Asteroideae and some Cichorioideae (globe artichoke, pot marigold, sunflower, yellow salsify) produced more Ψ-taraxasterol than taraxasterol. In contrast, enzymes from most species of the Cichorioideae either produced equal quantities of both products (endive, lettuce), or predominantly taraxasterol (Russian dandelion, Korean dandelion) (Supplementary Fig. 20). Using pot marigold and Russian dandelion as examples, we first investigated if this was reflected in the metabolic profile of plant extracts. Intriguingly, although this was found to be true, quantitative analysis also revealed that pot marigold accumulates Ψ-taraxasterol and derivatives (predominantly faradiol FAEs) exclusively in floral extracts, while Russian dandelion (Cichorioideae) predominantly accumulates taraxasterol roots with smaller quantities in the leaves and floral tissues (Fig. 5a, Supplementary Figs. 23–24).

To identify the residues involved in defining these highly similar triterpene scaffolds, and to investigate if there is evidence of natural selection on this enzyme, we investigated if the TXSSs producing predominantly taraxasterol that are found in all except the most basal groups of the Cichorioideae, are under positive selection (Fig. 5B; Supplementary Table 4). To do this, we constructed a maximum likelihood phylogenetic tree and fitted a branch-site model and a corresponding null model. This revealed that TXSSs in non-basal Cichorioideae are under positive selection (likelihood ratio test *p*-value < 0.01) and that 11 sites are likely to be under positive selection with probabilities higher than 0.9. Of these, two amino acid residues (D385 and H492; CoTXSS numbering) were within 12 Å of the predicted active site (Fig. 5C). In parallel, we aligned the sequences of all characterised TXSSs, identifying four residues within 12 Å of the active site that differed between non-basal Cichorieae TXSSs and sequences from other lineages: the two sites under positive selection (D385 and H492), G380 and P751 (Fig. 5B, C). To gain evidence that these residues influence product specificity, we made reciprocal mutants of CoTXSS and TkTXSS.

CoTXSS[D385E] and the reciprocal mutant TkTXSS[E379D] showed a significant reduction of both products indicating the importance of this residue (Fig. 5D, Supplementary Fig. 25; Supplementary Data 8). However, CoTXSS[D385E+H492Q] showed a reduction in Ψ-taraxasterol, resulting in a shift of the dominant product from Ψ-taraxasterol to taraxasterol. The mutation of the additional sites (those not under positive selection) CoTXSS[G380T+D385E+H492Q+P751A] had no further effect. In contrast, TkTXSS[E379D+Q486H] showed an increase in the production of Ψ-taraxasterol and a decrease in taraxasterol but only the quadruple mutant (TkTXSS[T374G+E379D+Q486H+A745P]) resulted in a reversal of the major product to Ψ-taraxasterol (Fig. 5D, Supplementary Fig. 25; Supplementary Data 8).

### Characterisation of a taraxasterol C16 hydroxylase

In pot marigold and field marigold, the majority of taraxasterol-based compounds are present as faradiol myristate and palmitate (Fig. 2 and Supplementary Table 1). Further, faradiol, arnidiol, and their fatty acid esters showed significant anti-inflammatory bioactivity (Fig. 2). To investigate the biosynthesis of these anti-inflammatory compounds, we identified candidate genes encoding cytochromes P450s (CYPs) able to catalyse the hydroxylation of Ψ-taraxasterol C16. Previously characterised CYPs active on C16 of other triterpene scaffolds include CYP51H10 (C12-C13β-epoxidase, C16β-hydroxylase) from *Avena strigosa* (black oat); AsCYP716 A111 (C16β-hydroxylase) from *Aquilegia coerulea* (rocky mountain columbine), and AcCYP716A141A (C28 oxidase, C16β-hydroxylase) from *Platycodon grandifloras* (Chinese bellflower). Due to its mono-functionality, AsCYP716A111 was used as a query to identify candidate genes in the pot marigold genome. Phylogenetic analysis of nine candidate genes from pot marigold, nine from field marigold, 68 uncharacterised candidate CYPs from publicly available genomes of Asteraceae, and 171 previously characterised CYPs confirmed that clades are predominantly grouped by substrate specificity (Supplementary Fig. 26). All nine pot marigold CYP candidates resolved in a clade containing previously characterised CYPs known to act on pentacyclic triterpene scaffolds including α-amyrin, β-amyrin and lupeol. Of these, five were identified in the transcriptome data and differential expression analyses indicated that *CoCYP716A392, CoCYP716A393, CoCYP716A429,* and *CoCYP716A430* were predominantly expressed in floral tissues, while *CoCYP716A431* was more highly expressed in leaves (Supplementary Fig. 27). The expression of the other genes may be limited to other tissues or specific conditions.

Candidates were characterised by co-expression with CoTXSS in *N. benthamiana* leaves. Co-expression of CoTXSS with either CoCYP716A392 or CoCYP716A393 resulted in the production of faradiol together with smaller quantities of maniladiol and calenduladiol indicating that these enzymes are C16β-hydroxylase able to use Ψ-

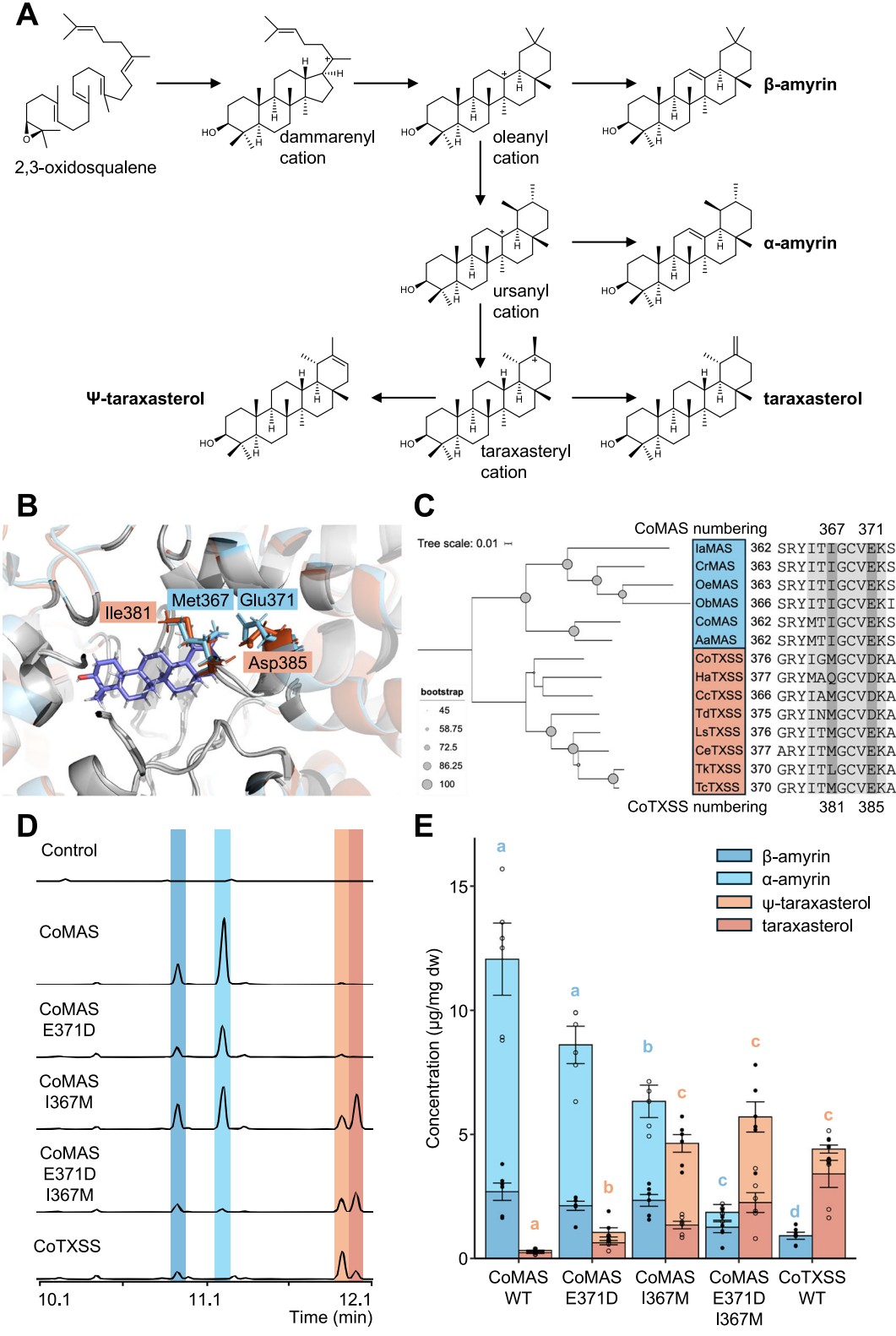

taraxasterol, β-amyrin and lupeol as substrates (Fig. 6A). CoCYP716A392 and CoCYP716A393 are 83.69% identical and encoded in genomic regions with conserved synteny indicating that they are likely homeologues (Supplementary Fig. 28). We note that the peak area of the terpene diols (faradiol, arnidiol, calenduladiol, and mani-ladiol) was not equivalent to the area depleted from the triterpene scaffold substrates. Notably, we did not detect the presence of arni-diol, despite a reduction in the taraxasterol peak, possibly the result of

derivation by an endogenous enzyme. Further, co-expression of CoTXSS with either CoCYP716A392 or CoCYP716A393 also resulted in a small quantity of faradiol palmitate, suggesting that an endogenous *N. benthamiana* acyl transferase (ACT) is able to add fatty acids to faradiol. Co-expression of CoTXSS with CoCYP716A431 resulted in the depletion of β-amyrin and lupeol; the presence of peaks corresponding to oleanolic and betulinic acids suggested that this enzyme is a promiscuous C28 hydroxylase (Supplementary Fig. 29). Co-expression

**Fig. 4 | Functional characterisation of a pot marigold Ψ-taraxasterol synthase (Co TXSS). A** Biosynthesis of Ψ-taraxasterol, taraxasterol, α-amyrin, β-amyrin and lupeol by 2,3-oxidosqualene cyclases. **B** Structural models showing the predicted position of the taraxasteryl cation in the active site of CoTXSS and CoMAS (mixed amyrin synthase). Residues that differ are highlighted in blue. **C** Phylogenetic tree and sequence alignment of selected residues of MASs and TXSSs with active site residues highlighted in grey and residues selected for mutagenesis in dark grey. Ia *Ilex asprella;* Cr *Catharanthus roseus;* Oe *Olea europaea;* Ob *Ocimum basilicum;* Co *Calendula officinalis;* Aa *Artemisia annua;* Ha *Helianthus annuus;* Cc *Cynara cardunculus;* Td *Tragopogon dubius;* Ls *Lactuca sativa;* Ce *Cichorium endivia;* Tk *Taraxacum kok-saghyz;* Tc *Taraxacum coreanum.* A list of taxa and accession numbers/ protein sequences are provided in Supplementary Table 4. **D** Total ion chromatograms of extracts of *N. benthamiana* leaves transiently expressing wild type and mutated CoMAS and CoTXSS. **E** Quantification of triterpenes produced by wild type and mutated CoMAS and CoTXSS. Error bars indicate the mean and standard error of 6 biological replicates (independent infiltrations). Significant differences in α/β-amyrin content (blue lowercase letters) and Ψ-taraxasterol/taraxasterol content (red lowercase letters) were analysed using a Kruskal-Wallis test followed by post-hoc Wilcoxon rank sum test with a Benjamini-Hochberg correction (Supplementary Data 7). Samples that do not share the same lower-case letter are significantly different from each other ($p < 0.05$). $P$ values are provided in Supplementary Data 7.

---

of CoTXSS with Co CYP716A429 or CoCYP716A430 did not yield any new peaks. In addition, we tested the activity of a candidate CYP from field marigold, that was closely related to CoCYP716A392 or CoCYP716A393 and also catalysed the production of faradiol (Supplementary Fig. 29). The clade containing CYPs active on pentacyclic triterpenes also contained sequences from common sunflower (Supplementary Fig. 26). To investigate if this species accumulates faradiol, we analysed floral extracts identifying low quantities (as compared to pot marigold), which are likely insufficient to confer detectable anti-inflammatory activity (Supplementary Fig. 30).

To identify residues important for activity on Ψ-taraxasterol, we examined the structure of the predicted active sites CoCYP716A392 and CoCYP716A393. To do this, we inferred structural models using AlphaFold2[24], which showed high similarity (RMSD: 1.283 CoCYP716A392; RMSD:1.235 CoCYP716A393) to the crystal structure of CYP90B1A in a complex with cholesterol (PDB:6A15)[26]. Based on the position of cholesterol in CYP90B1A, we predicted residues within 12 Å of the active site (Fig. 6B). We compared these residues in 22 CYPs active on β-amyrin or Ψ-taraxasterol, including the C:16 hydroxylases, CoCYP716 A111 and CoCYP716A141, and two CYPs from pot marigold and two CYPs from field marigold. Three sites, A285, A357, and H424, within an otherwise conserved region were observed to differ between β-amyrin and Ψ-taraxasterol hydroxylases. These were mutated in both pot marigold enzymes, CoCYP716A392 and CoCYP716A393, and the mutants were co-expressed in *N. benthamiana* with CoTXSS. As previously observed when expressing the wild-type CYPs (Fig. 6A), the quantity of diols did not correspond to the reduction of the substrate peak. We therefore quantified the reduction of the substrates (Ψ-taraxasterol/ taraxasterol and β-amyrin). In both enzymes, the A285V mutation resulted in a greater depletion of β-amyrin and reduced depletion of Ψ-taraxasterol. In CoCYP716A392 the A357L and H424R mutations also shifted activity towards β-amyrin (Fig. 6C, D; Supplementary Fig. 31; Supplementary Data 9).

## Reconstruction of TFAE biosynthesis in *N. benthamiana*

We next sought to identify enzymes capable of adding fatty acid groups to Ψ-taraxasterol/ taraxasterol and/or faradiol to enable pathway reconstruction of pot marigold triterpene fatty acid esters. To do this, we used *Arabidopsis thaliana* thalianol acyl transferase 3 (AtTHAA3; AtASAT1; At3g51970), as a query to search the transcriptome and genome of pot marigold and publicly available Asteraceae species. We identified 13 candidate pot marigold triterpene *ACT* genes (*CoACTs 1-13*), predicted to be members of the membrane-bound O-acyltransferase (MBOAT) superfamily, two of which contained missense mutations (Supplementary Fig. 32). As observed for CoOSCs and CoCYPs, many candidate proteins were paired in the phylogeny. In these cases, we selected one of each pair for further study (*CoACT1-7*). *CoACT1-3* were all more highly expressed in flowers than leaves, with *CoACT1* and *CoACT2* upregulated in ray florets compared to disk florets (Supplementary Fig. 33). All seven candidates were cloned and co-expressed with constructs expressing either *CoTXSS* alone, or *CoTXSS* and *CoCYP716A392* in *N. benthamiana*

(Fig. 7A; Supplementary Figs. 34–35). As noted above, small quantities of faradiol palmitate were detected in *N. benthamiana* leaves infiltrated with constructs expressing *CoTXSS* and *CoCYP716A392*. Therefore, samples expressing candidate *CoACTs* were compared to control samples expressing these genes. In samples co-infiltrated with *CoTXSS* alone, samples with all *CoACTs* except *CoACT1* and *CoACT6* accumulated more taraxasterol/Ψ-taraxasterol palmitate than controls indicating activity on both triterpene scaffolds (Fig. 7B; Supplementary Fig. 34; Supplementary Data 10). Samples infiltrated with constructs expressing *CoACT3* produced the largest quantities of Ψ-taraxasterol/taraxasterol palmitate (Fig. 7B; Supplementary Data 10). When co-expressed with *CoTXSS* and *CoCYP716A392*, only *CoACT1* and *CoACT2* produced more faradiol palmitate than the control. *CoACT1* and *CoACT2* are encoded in genomic regions with conserved synteny indicating that they are likely homeologues (Supplementary Fig. 36).

Having identified genes for the production of faradiol FAEs (*CoTXSS, CoCYP716A392, CoCYP716A393, CoACT1 and CoACT2;* Fig. 7C), we noted that the genes for each pathway step are not co-located in the genome. To investigate if their expression was synchronous, we extracted RNA from flowers sampled at six stages of floral development (Fig. 7D) and compared the expression of each gene by qRT-PCR. We found the expression to be asynchronous, with *CoTXSS, CoACT1 and CoACT2*, being most highly expressed in young buds with limited expression later in development and *CoCYP716A392, CoCYP716A393* being most highly expressed in mature buds (Fig. 7D). Metabolic profiling of samples from the same flowers indicated that the abundance of Ψ-taraxasterol and fardiol palmitate is low in buds and increases though floral development reaching 8.6 µg/mg and 15.9 µg/mg, respectively in the oldest flowers (Supplementary Fig. 37). As observed in the metabolic profile of mature flowers (Supplementary Table 1), the abundance of the pathway intermediate, faradiol, remained very low throughout development, but was slightly higher in developing buds (Supplementary Fig. 37).

## Discussion

The development of drugs from natural products requires both the identification of specific compounds and a route to access sufficient quantities of those molecules. Plants have historically been an important source of drugs[27,28]. However, there are numerous technical challenges to the isolation and characterisation of bioactives. Even when a specific compound has been identified, it may occur in complex mixtures or accumulate at low volumes in limited cell types. These challenges, together with rapid advances in chemical synthesis in 20th century led to a decline in the importance of plants as sources of new drugs[29].

In recent decades, a rapid increase in plant genome information and powerful bioinformatic tools for genome and protein analysis have made pathway discovery more accessible[30,31]. Alongside this, advances in molecular and synthetic biology have made the functional characterisation of candidate enzymes increasingly easy. Notably, the plant system *N. benthamiana*, has become a well-used experimental platform for the characterisation of plant enzymes and the

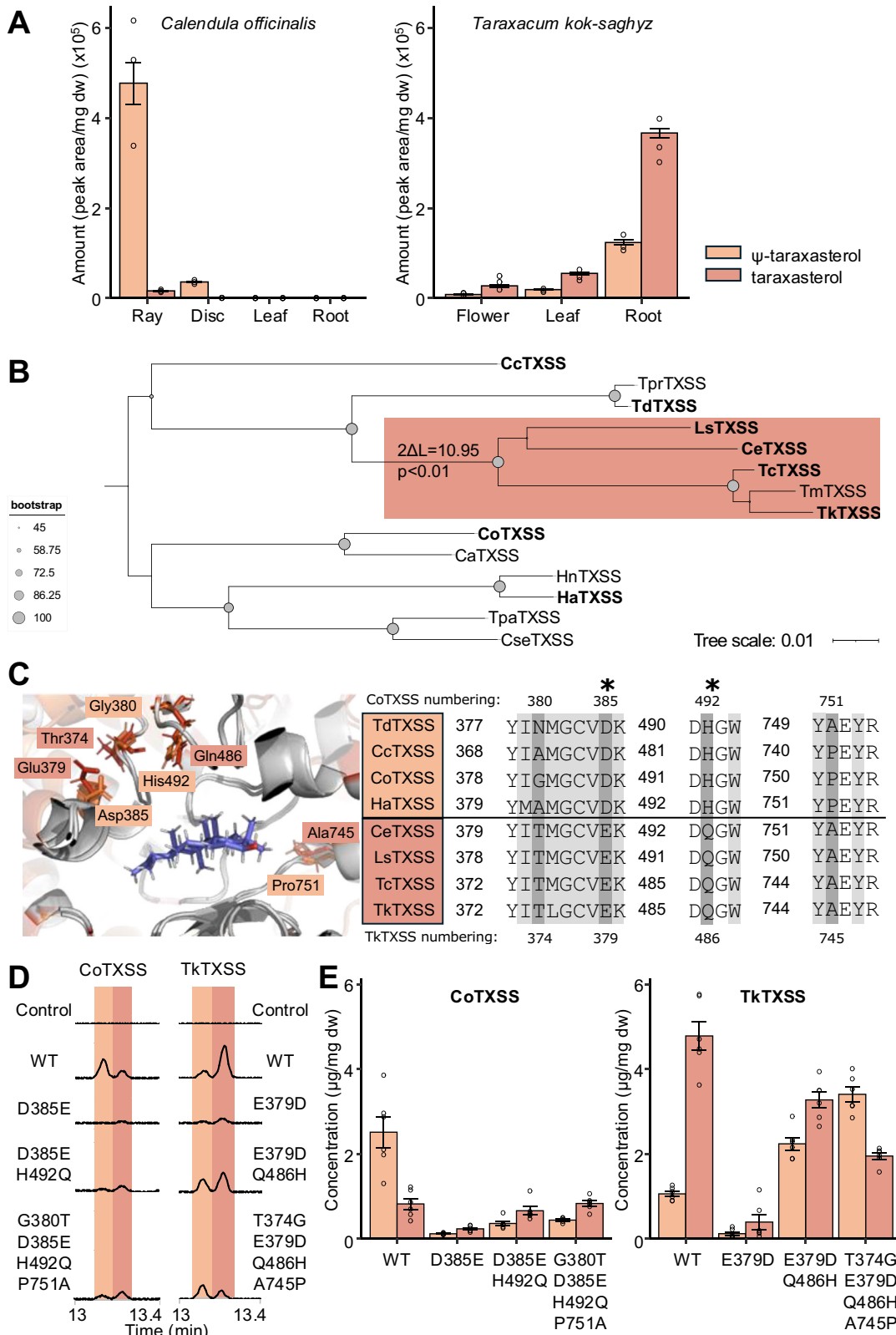

reconstruction of plant natural product pathways[32]. This ability to program easily cultivable organisms to produce high levels of natural products has the potential to provide sustainable routes to access of plant natural products[32–34].

While molecules responsible for the bioactivity of some species used in traditional medicines have been definitively identified, in many species, compounds remain unknown. In this study, we simultaneously investigated the identity of bioactive molecules found in pot marigold and elucidated the genetic basis of their production. This enabled us to confirm one of the molecules responsible for anti-inflammatory bioactivity (Fig. 2D) and understand why similar bioactivity has not been reported for other species that accumulate closely related molecules. It also enabled us to reconstruct both biosynthesis of faradiol palmitate and that of the highly bioactive intermediate, faradiol.

**Fig. 5 | Identification of residues involved in the product specificity of TXSSs.**
**A** Peak area analysis of triterpenoids detected in the floral, leaf and root tissues of pot marigold and Russian dandelion plants. Error bars indicate the mean and standard error of 3 biological replicates. **B** Phylogenetic tree of TXSSs in which the foreground branch containing taxa that predominantly produce taraxasterol is highlighted in light red. The Log likelihood ratio test at the foreground branch node indicates that this branch is likely under positive selection ($p < 0.01$). Cc *Cynara cardunculus*; Tpr *Tragopogon pratensis*; Td *Tragopogon dubius*; Ls *Lactuca sativa*; Ce *Cichorium endivia*; Tc *Taraxacum coreanum*; Tm *Taraxacum mongolicum*; Tk *Taraxacum kok-saghyz*; Co *Calendula officinalis*; Ca *Calendula arvensis*; Hn *Helianthus niveus*; Ha *Helianthus annuus*; Tpa *Tanacetum parthemium*; Cse *Chrysanthemum seticuspe*. Accession numbers are provided in Supplementary Table 4. **C** Structural models illustrating the predicted position of the taraxasteryl cation in the active site of CoTXSS and TkTXSS. Residues that differ between them are highlighted in orange (CoTXSS) and red (TkTXSS). Asterisks indicate that residues are under positive selection in non-basal Cichorieae **D** Total ion chromatograms of extracts of *N. benthamiana* leaves transiently expressing wild type and mutated CoTXSS and TkTXSS; **E** Quantification of triterpenes produced by transiently expressing wild type and mutated CoTXSS and TkTXSS; Error bars indicate the mean and standard error of 6 biological replicates (independent infiltrations). Significant differences in Ψ-taraxasterol (orange lowercase letters) and taraxasterol (red lowercase letters) were analysed using a Kruskal-Wallis test followed by post-hoc Wilcoxon rank sum test with a Benjamini-Hochberg correction. Samples that do not share the same lower-case letter are significantly different from each other ($p < 0.05$). *P* values are provided in Supplementary Data 8.

For the latter, we obtained yields of up to 2.34 µg/mg dw (Fig. 7), 9.4-fold higher than those found in extracts of mature pot marigold flowers (0.25 µg/mg dw; Supplementary Fig. 37) and 3.8-fold higher than those found in developing buds (0.61 µg/mg dw; Supplementary Fig. 37) (Supplementary table 1). We note that these are the initial yields of pathway reconstruction. Future metabolic engineering efforts are required to optimise yields and to identify the best methods and chassis in which to produce these compounds.

While our work confirmed that faradiol and its fatty acid ester are key contributors to anti-inflammatory activity of pot marigold extracts (Fig. 2D), in agreement with previous observations[15,17], our experiments with fractionated extracts indicate that other molecules are likely to contribute to the overall activity (Fig. 2C). Specifically, while the fraction containing faradiol FAEs displayed significant anti-inflammatory activity via the IL-6 pathway, fractions containing fatty acids also showed significant bioactivity. Also consistent with the literature, we observed that triterpene diols faradiol and arnidiol have the most significant anti-inflammatory bioactivity (Fig. 2D)[12,13].

Our investigation of the mechanisms by which faradiol influences IL-6 production in LPS-stimulated monocytes revealed an unexpected result: inhibiting phosphorylation of STAT3 (Fig. 2E). A recent investigation of the role of long non-coding RNA brain and reproductive organ-expressed protein (BRE) antisense RNA 1 (BRE-AS1) as a regulatory element in the LPS-induced JAK2/STAT3 inflammatory pathway showed that knockdown of BRE-AS1 enhances LPS-induced expression of IL-6 and IL-1β, but does not affect levels of TNF-α[35]. These results suggest that induction and regulation of JAK2/STAT3 is independent of the TNF-α pathway. In previous work, faradiol was reported to inhibit NF-κB-driven transcription of a synthetic reporter gene in AGS (adenocarcinoma of the stomach) cells[17]. Here, it did not inhibit p65 NF-κB phosphorylation, which may not be required for TNF or IL6 transcriptional activation in response to LPS. More work is needed to clarify if NF-κB is influenced by faradiol and, if so, which part of the pathway is involved and to what extent these previously reported effects are cell-specific.

Taraxasterol and psi-taraxasterol are found as esters, acetates, diols and triols across the Asteraceae[36]. However, we identified only a few species that are likely to be capable of producing faradiol, as determined by the presence of orthologs of CoCYP716A392 and CoCYP716A393 (Fig. 6; Supplementary Fig. 26). While faradiol compounds are abundant in *Calendula* species, other species were found to accumulate relatively small quantities (Supplementary Fig. 30), perhaps explaining the use of this genus in wound healing remedies. We also found that the closely related compound, arnidiol, to have comparable activity to faradiol (Fig. 2D). Interestingly, arnidiol takes its name from the species from which it was first isolated, *Arnica montana* L. (wolf's bane), extracts of which are used in topical treatments for bruising and muscle pain. Decorations of the C16 position of triterpenes have also been associated with antimicrobial bioactivity, toxicity and the ability to suppress neuroinflammation[37-39].

Our ability to characterise the CYPs was likely compromised by the endogenous metabolism of *N. benthamiana*: yields of faradiol were not equivalent to the reduction in Ψ-taraxasterol substrate and we were unable to detect arnidiol in this species, despite a reduction in the presence of the taraxasterol substrate (Fig. 6C, D). It is most likely that these products were derivatised by an endogenous enzyme as we and others have previously reported for a range of other molecules[40-45]. We also note that in pot marigold we detected the presence of faradiol-stearate, -myristate and -palmitate but expression of all pathway genes in *N. benthamiana*, only resulted in the accumulation of faradiol palmitate (Fig. 7). This is likely due to the availability of fatty acids in this species, although further characterisation of the ACTs is required to confirm this.

Previously, TXSS genes have been identified and characterised from Cichorioideae species, Russian and Korean dandelion, and lettuce. Here we characterised six TXSSs, finding that they all are multifunctional (Supplementary Fig. 20). Multifunctionality has been described for numerous triterpene-producing OSCs[19]. However, as little is known about the biological function of most plant triterpenes, or at which concentration they are bioactive, it is not possible to predict if these are side-products of promiscuous enzymes or if these production profiles have biological significance[46,47]. Interestingly, we found evidence that Cichorioideae TXSSs show evidence of positive selection (Fig. 5B). Coupled with our observed differences in the site of accumulation (Fig. 5A), this suggests that taraxasterol and Ψ-taraxasterol may fulfil different biological functions in plants. Further, we observed differences in the bioactivity of Ψ-taraxasterol compared to taraxasterol (Fig. 2D).

The multifunctionality of triterpene-producing OSCs is also likely responsible for trace quantities of taraxasterol reported in non-Asteraceae species e.g. olive, Arabidopsis, pea and tomato[48-51]. We found no evidence of TXSS genes outside the Asteraceae (Fig. 3) and Bayesian analysis suggested that TXSSs emerged 83-121 mya (Supplementary Fig. 21), consistent with duplication after divergence of Asteraceae, estimated to have occurred 64-91 mya[1].

Further, knowledge of the relationship between structure and product specificity of OSC can be applied to engineering[52]. We identified two residues important for a decrease in the production of β-amyrin and an increase in production of Ψ-taraxasterol (Fig. 4), as well as residues that control the ratio of Ψ-taraxasterol:taraxasterol (Fig. 5).

Interestingly, though genes for some plant terpenes are found to be clustered within plant genomes and/or tightly co-expressed[53], although pathway gene expression was confined to floral tissues (Supplementary Figs. 19, 27 and 33), the genes of this pathway are not co-located, and expression levels differed through floral development (Fig. 7D), which is consistent with the accumulation of pathway intermediates (Fig. 2A, Supplementary Fig. 1, Supplementary Table 1). While we detected evidence of homeologous pairs for all pathway genes, one copy of the TXSS contained mutations indicative of pseudogenisation (Supplementary Figs. 18, 28 and 36), indicative of diploidisation seen in ancient polyploids[54].

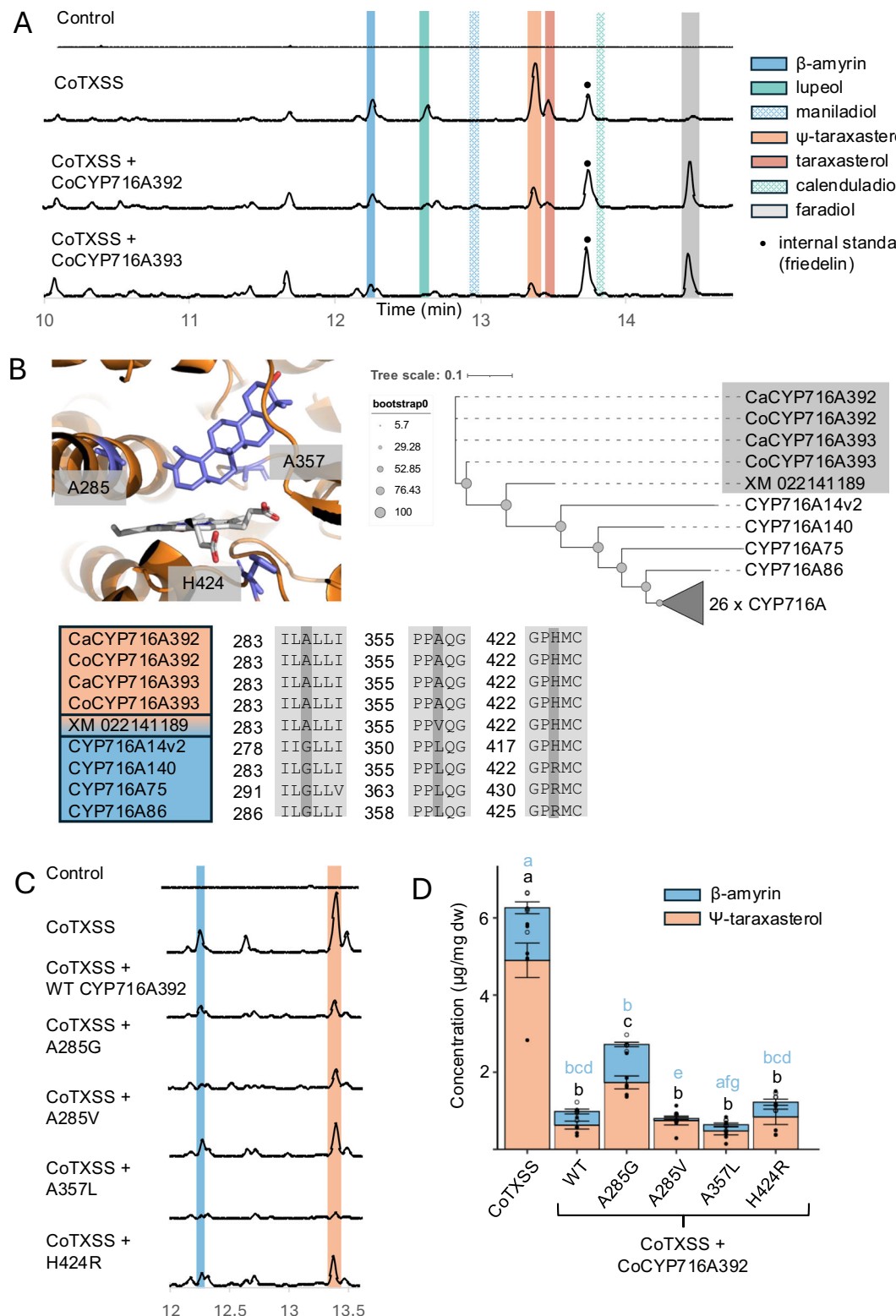

Pathway elucidation and reconstruction in heterologous hosts provides opportunities for sustainable access to useful molecules such as faradiol, which we have shown exhibits anti-inflammatory activity by preventing the phosphorylation of STAT3. The ability to use metabolic profiling of plant extracts to simultaneously investigate bioactivity and elucidate the biosynthetic pathway has the potential to fast-track the discovery and production of bioactive plant natural products.

## Methods

### Plant growth

Seeds of *Calendula officinalis* (pot marigold) and *Helianthus annuus* (common sunflower) were obtained from Chiltern Seeds (Wallingford, UK). Seeds of *Calendula arvensis* (field marigold; # 32133) were obtained from the Millennium Seed Bank (Wakehurst, UK). Seeds of *Taraxacum kok-saghyz* (Russian dandelion; #W635156) were obtained

**Fig. 6 | Characterisation of pot marigold C16 hydroxylases. A** Total ion chromatograms of extracts of *N. benthamiana* leaves transiently co-expressing CoTXSS with CoCYP716A392 or CoCYP716A393. Ca *Calendula arvensis* (field marigold); Co *Calendula officinalis* (pot marigold). **B** Structural model illustrating the predicted position of Ψ-taraxasterol in the active site of CoCYP716A392. Residues selected for mutagenesis are highlighted in grey. Phylogenetic tree of CYP716A and alignment of residues in the active sites; taxa that are predominantly use Ψ-taraxasterol as a substrate are highlighted in orange and those use β-amyrin are highlighted in blue. The grey triangle represents a collapsed clade of 26 characterised CYPs from the CYP716A subfamily that hydroxylate residues of β-amyrin other than C16.
**C** Extracted ion chromatograms and peak area analysis of extracts of *N. benthamiana* leaves transiently co-expressing CoTXSS with wild type and mutated CoCYP716A392. Control = HMGR + P19. **D** Quantification of triterpenes produced by transiently expressing CoTXSS with wild type and mutated CoCYP716A392. Error bars indicate the mean and standard error of 6 biological replicates (independent infiltrations). Significant differences in total Ψ-taraxasterol/β-amyrin content compared to wild type CoCYP716A392 (black lowercase letters), and significant differences in taraxasterol/β-amyrin ratio compared to wild type CoCYP716A392 (blue lowercase letters) were analysed using a Kruskal-Wallis test followed by post-hoc Wilcoxon rank sum test with a Benjamini-Hochberg correction (Supplementary Data 7). Samples that do not share the same lower-case letter are significantly different from each other ($p < 0.05$). *P* values are provided in Supplementary Data 9.

from the National Plant Germplasm System Germplasm Resources Information Network (GRIN). Seeds were sown in 9 cm plastic pots containing Levington F2 starter (100% peat). Approximately ten days post-emergence, seedlings were transplanted into 11 cm plastic pots containing John Innes cereal mix (60% peat, 20% grit, 20% perlite, 2.25 kg/m3 dolomitic limestone, 1.3 kg/m3 PG mix, 3 kg/m3 Osmocote Exact). Plants for genome sequencing and transcriptomics were grown in controlled environment conditions at 22 °C (day) and 25 °C (night) with a 16-h photoperiod. All other plants were grown in summer glasshouse conditions with natural day length and temperature. In addition, Russian dandelion seedlings were cold treated at 4 °C for four weeks before returning to summer glasshouse conditions. *Nicotiana benthamiana* was cultivated in a peat-based potting mix (90% peat, 10% grit, with 4 kg/m3 dolomitic limestone, 0.75 kg/m3 powdered compound fertiliser, 1.5 kg/m3 slow-release fertiliser). *N. benthamiana* plants were grown in a controlled environment room with 16 h light, 8 h dark at 22 °C, 80% humidity and ~200 μmol/m2/s light intensity.

### Metabolite extraction and GC-MS analysis
For analysis of Asteraceae species, 30 mg of freeze-dried plant material was homogenised using 3 mm tungsten carbide beads (Qiagen, Hilden, Germany) with a TissueLyser (25 Hz, 1 min) and 500 μL ethyl acetate was added (Sigma Aldrich, Burlington, MA, USA). For *N. benthamiana*, one to five freeze-dried discs (1 cm) were sampled from infiltrated leaves and similarly homogenised. 100 μL ethyl acetate (Sigma Aldrich) was added per 2 mg of dry weight. Samples were agitated in ethyl acetate at 700 rpm at 40 °C for 2 h then at room temperature for 48 h. Plant material was collected by centrifugation at $21,300 \times g$ for 5 min and 50 μL of supernatant was transferred to a 2 mL glass vial. Samples were dried in a centrifugal evaporator and derivatised with 50 μL N-Methyl-N-(trimethylsilyl)trifluoroacetamide (Sigma Aldrich) at 37 °C for 30 min then transferred to glass inserts in 2 mL glass vials.

GC-MS analysis was performed using a 7890B GC (Agilent; Santa Clara, CA, USA) fitted with a Zebron ZB5-HT Inferno column (Phenomenex; Washington, D.C, USA). Injections (2 μL) were performed in pulsed splitless mode (10 psi pulse pressure) with the inlet temperature set to 325 °C. The GC oven temperature program was 150 °C and held for 30 seconds with subsequent increase to 360 °C (20 °C/min) and held at 360 °C for an additional 12.5 min (total run time 27 min). The GC oven was coupled to an Agilent 5977B Mass Selective Detector set to scan mode from 60-800 mass units (solvent delay 3 min). Data analysis was carried out using MassHunter workstation software version B.08.00 (Agilent).

### LC-MS analysis
For analysis of *Calendula officinalis*, 10 mg of freeze-dried plant material was homogenised using 3 mm tungsten carbide beads (Qiagen, Hilden, Germany) with a TissueLyser (25 Hz, 1 min) and 500 μL 80% methanol was added (Sigma Aldrich, Burlington, MA, USA). Samples were agitated in ethyl acetate at 700 rpm at 40 °C for 20 min. Plant material was collected by centrifugation at 21,300 x g for 5 min

and the supernatant was transferred to a 2 mL glass vial. Samples were dried in the fume hood at room temperature for 3 days. Samples were resuspended in 100 μL 80% methanol, filtered through 0.2 μm Mini-Filter Spin Columns (Geneflow; Staffordshire, UK) then transferred to glass inserts in 2 mL glass vials.

LC-MS analysis was performed using a 6546 LC/Q-TOF (Agilent; Sata Clara, CA, USA) fitted with a 1.7 μM Acuity UPLC BEH C18 column (Waters; Wilmslow, UK). Separation was carried out using 0.1% formic acid in water (A) versus acetonitrile (B) run at 0.6 mL/min and following gradients of solvent B; 15% from 0–0.75 min, 15–60% from 0.75–13 min, 60–100% from 13–13.25 min, 100–15% from 13.25–14.5 min, and 15% 14.5–16.5 min. Analytes were detected by negative electrospray ionisation using the JetStream source. The instrument collected full spectra from *m/z* 100–1700 (200msec per spectrum), and data-dependent MS/MS spectra for the two most abundant precursors (125msec per spectrum) with medium isolation width (*m/z* 4) and 35% collision energy. Spray chamber conditions were 10 L.min$^{-1}$ drying gas at 325 °C, 20 psi nebulizer pressure, 12 L.min$^{-1}$ sheath gas at 400 °C, 120 V fragmentor voltage, 3500 V Vcap, and 1000 V nozzle voltage. Data analysis was carried out using MassHunter workstation software version B.08.00 (Agilent).

### Fractionation and purification of floral extracts
Extracts were fractionated using an adaptation of a previously described method[55]. Liquid chromatography of methanol extracts was performed on an ACQUITY UPLC BEH C18 2.1 mm × 50 mm column (Waters Corp.; Milford, MA, USA) on a single quadrupole LC-MS/MS (Nexera UHPLC from Shimadzu; Kyoto, Japan). The flow rate was set to 0.6 ml/min and the column temperature was kept constant at 40 °C for 28 min. Eluent A (50% methanol) was applied for 2.5 min followed by a gradient of 85% to 100% methanol for 20 min, followed by eluent B (100% methanol) for 2.5 min. Sequential fractions of 1.8 ml were collected and dried to yield seven samples of 234 μg, 253 μg, 213 μg, 325 μg, 244 μg, 593 μg, and 237 μg. These fractions were analysed by GC-MS before use in cell proliferation and cytokine assays. To purify faradiol palmitate the same method was used except that eluent A (90% methanol) was applied for 2.5 min followed by a gradient of 90% to 97.5% methanol for 20 min, followed by eluent B (97.5% methanol) for 5.5 min. Sixteen fractions were collected and analysed by GC-MS to identify a fraction containing a single peak with the mass and spectra for faradiol palmitate (15-16 min). This fraction was collected and dried, yielding 1.2 mg of compound.

### Nuclear magnetic resonance (NMR) spectroscopy
The structure of faradiol palmitate was confirmed by 1D and 2D NMR analysis. Spectra were recorded in 3 mm tubes using CDCl3 as a solvent at 298 K on a Bruker Neo 600 MHz spectrometer (Billerica, MA, USA) equipped with 5 mm TCI CryoProbe. 1D 1H, 13 C NMR, 2D 1H-1H-COSY, 1H-13C-HSQCed and 1H-13C-HMBC experiments were performed using standard pulse sequences from the Bruker Topspin 4 library. Data was analysed using Topspin 4.1.4 and MestReNova 15.0.1 software and spectra were calibrated to an internal TMS reference.

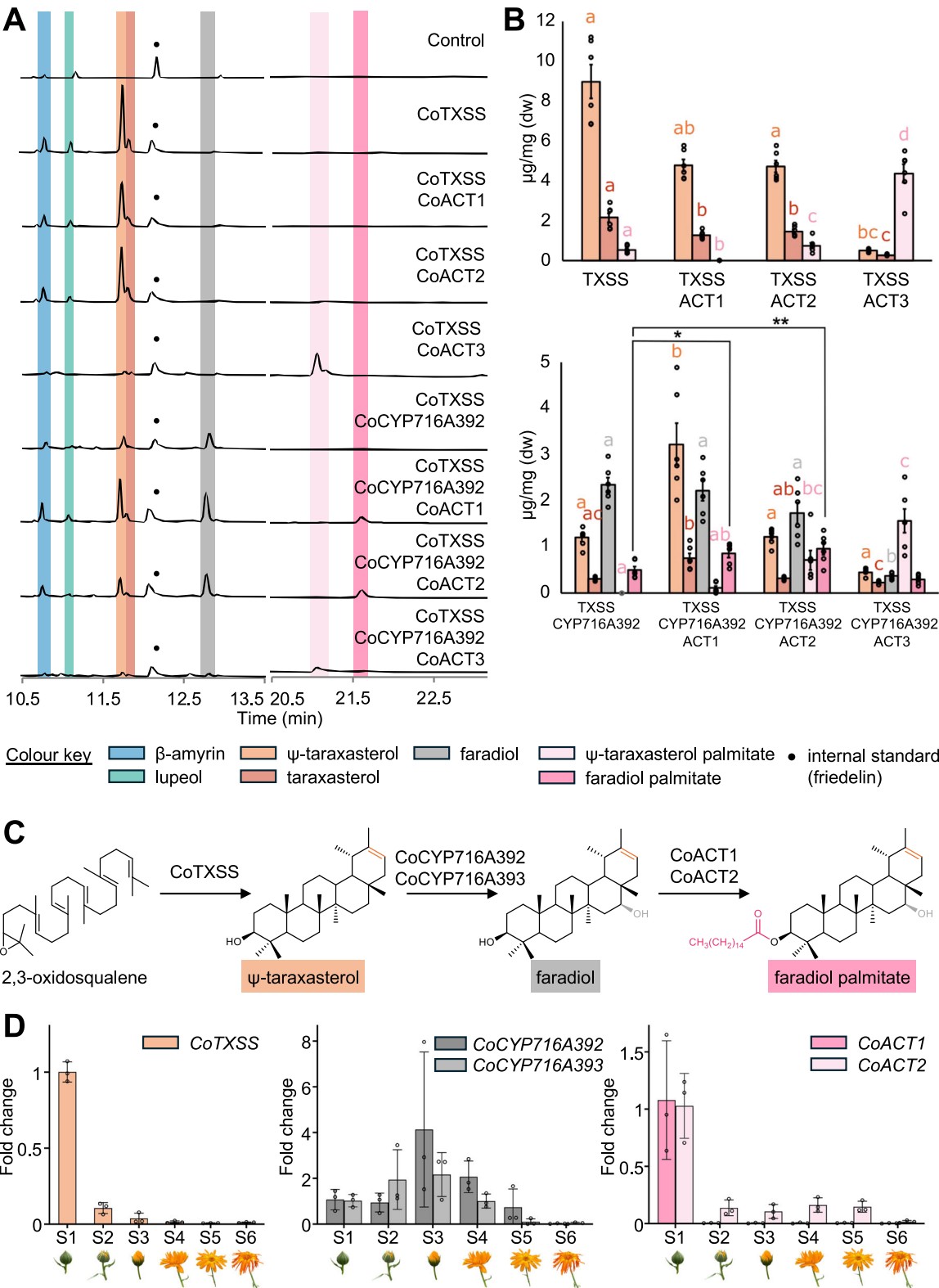

**Fig. 7 | Characterisation of pot marigold triterpene acyl transferases, pathway reconstruction and expression analysis. A** Total ion chromatograms of extracts of *N. benthamiana* leaves transiently co-expressing CoTXSS with CoACT1-3 or CoTXSS, CoCYP716A392 and CoACT1-3. **B** Peak area analysis showing products of pathway reconstruction in *N. benthamiana* through agroinfiltration of indicated genes, quantified compared to friedelin internal standard. Error bars indicate the mean and standard error of 6 biological replicates (independent infiltrations).

Statistical significance was inferred using a Kruskal-Wallis test with a post-hoc Dunns test and Benjamini-Hochberg correction, except faradiol palmitate which was inferred by one-way ANOVA and post-hoc Dunnetts test (Supplementary Data 10). **C** Pathway schematic. *P* values are provided in Supplementary Data 10. **D** Relative expression of *CoTXSS, CoCYP716A392, CoCYP716A393, CoACT1, CoACT2* through six stages of floral development (S1-6); Error bars indicate the mean and standard deviation of 3 biological replicates (2 technical replicates per sample).

## Cell maintenance and proliferation assays

The human monocytic leukaemia cell line THP-1 (ECACC 88081201) was obtained from the European Collection of Cell Cultures (Health Protection Agency, Salisbury, UK). THP-1 cells were cultured in Roswell Park Memorial Institute (RPMI) 1640 media supplemented with 10% heat-inactivated bovine foetal calf serum (FCS), L-glutamine (2 mM) and antibiotics (penicillin (100 U/mL); streptomycin (100 μg/mL) (GIBCO, Carlsbad, CA, USA). Cells were maintained at 37 °C in a humidified atmosphere with 5% $CO_2$ and passaged every 3.5 days to ensure the desired cell density of $3 \times 10^5$ to $9 \times 10^5$ cells/mL. Cell density was measured using a Neubauer haemocytometer according to the manufacturer's instructions (Sigma-Aldrich, Poole, UK). THP-1 cells were seeded into a 96-well plate at a density of $1 \times 10^6$ cells/mL in 100 μL volume. The appropriate wells were left untreated or treated with solvent control (DMSO; Sigma-Aldrich) and the compounds of interest as previously described for 24 h at 37 °C/5% $CO_2$[56]. 3-(4,5-dimethylthiazol-2-yl)-5-(3-carboxymethoxyphenyl)-2-(4-sulfophenyl)-2H-tetrazolium MTS (CellTiter 96 Aqueous One Solution Reagent, Promega, Southampton, UK) was added for 3 hrs at 37 °C; 5% $CO_2$ and absorbance at 492 nm was measured using a POLARstar Optima microplate reader (BMG Labtech, Aylesbury, UK). Data analysis was performed using GraphPad Prism software (v. 10.3.1) (Dotmatics, Boston, MA, USA).

## Cytokine assays

THP-1 cells were seeded into a 24-well plate at $1 \times 10^6$ cells/ml and treated with triterpenoids or plant extracts at predetermined non-cytotoxic concentrations of 20 μM and 50 μg/ml, respectively. DMSO was used as a solvent control. BAY 11-7082, a NLRP3 inflammasome and NF-κB inhibitor (E)−3-(4-Methylphenylsulfonyl)−2-propenenitrile (BAY; 10 μM, Sigma-Aldrich), was used as a positive control. Cells were incubated at 37 °C/5% $CO_2$ for 30 min prior to treatment with 10 ng/mL LPS from *Escherichia coli* O55:B5 (Sigma Aldrich) for 3 hrs to stimulate TNF-α release or 1 μg/ml LPS for 24 h to stimulate IL-6 release. Supernatants were then collected and stored at -80 °C until required. TNF-α or IL-6 concentrations were determined using the OptEIA human TNF-α ELISA and human Il-6 ELISA sets (BD Biosciences, Berkshire, UK), according to the manufacturer's instructions.

## Western blotting

To extract protein, THP-1 cells were lysed with 1:1 Novex tris-glycine sodium dodecyl sulphate lysis buffer (Thermofisher Scientific, Loughborough, UK)/PBS then boiled and sonicated. Proteins were quantified and separated by electrophoresis on a 4-12% NuPAGE bis-tris gel (Thermo Fisher Scientific) and transferred to a polyvinylidene difluoride membrane using the XCell SureLock™ Mini-Cell and XCell II™ Blot Module (Thermo Fisher Scientific). The membrane was blocked with 5% dried skimmed milk in tris-buffered saline with 0.1% Tween (blocking buffer) overnight at 4 °C before adding the primary antibody for 1.5 h at room temperature. Primary rabbit monoclonal antibodies against human STAT3 (#12640, 1:1000), phospho-STAT3 (Tyr705) (#9145, 1:1000), NF-κB p65 (#8242, 1:500), phospho-NF-κB p65 (Ser536) (#3033, 1:750), and β-Tubulin (#2128, 1:1000) were obtained from Cell Signalling Technology (London, UK). After stringent washing in blocking buffer and TBST, the membrane was incubated with the goat anti-rabbit IgG secondary antibody conjugated with horseradish peroxidase (HRP) in blocking buffer (#7074, 1:1000) (Cell Signalling Technology) for 40 min. After washing, chemiluminescence was detected using Invitrogen ECL-HRP substrate reagent (Thermo Fisher Scientific) and the Image Quant LAS 4000 (GE HealthCare, Chalfont St Giles, UK).

## Purification of nucleic acids

Leaves were harvested from young (16-18 days post germination) pot marigold plants. Flowers were harvested from mature (flowering) plants and deconstructed to separate ray florets and disk florets. Plant tissues were flash frozen in liquid nitrogen and stored at −80 °C. High molecular weight DNA was purified from young leaf tissue using the Nucleon Phytopure kit (Cytiva, Portsmouth, UK) according to the manufacturer's instructions. DNA quantity and size distribution were measured using a Bioanalyzer (Agilent Technologies) and Femto Pulse (Agilent Technologies).

Total RNA was purified from leaf, disc floret and ray floret tissue collected from four independent plants, snap-frozen frozen in liquid nitrogen and stored at −70 °C. RNA was extracted using the Spectrum Plant Total RNA Kit (STRN250; Merck, Darmstadt, Germany) following manufacturer's instructions with the following exception: during column washing, 300 μl of column wash 1 was used to wash RNA and then 80 μL of DNase I (RNA-Free Dnase Set 79254; Qiagen, Hilden, Germany) was added to the column and incubated for 15 min followed by 500 μl of wash 1. RNA integrity was assessed by gel electrophoresis on a 1.5 % agarose gel and RNA quantity and size distribution were measured using a Bioanalyzer (Agilent Technologies, Sata Clara, CA, USA).

## Genome assembly and differential expression analyses

Pot marigold Genomic DNA was sequenced using Illumina, PacBio, Chromium linked-read, and OmniC technologies (see Supplementary Methods). Pot marigold and field marigold cDNA were sequenced using Illumina and PacBio (Isoseq) chemistries. The genome and transcriptome were assembled and annotated as detailed in the extended methods (see Supplementary Methods). Contig synteny was determined using the progressive Mauve algorithm[57] and visualised in Geneious Prime® 2024.0.5 (https://www.geneious.com). Expression was quantified with salmon quant and differential gene expression analysis was carried out using DESeq2 (R: 3.6.2; DESeq2: 1.26.0). Conditions were compared pairwise as follows: Disc vs Leaf; Disc vs Ray; Leaf vs Ray. For comparisons involving disc tissue samples, "disc" was set as the baseline, while leaf was set as baseline for leaf vs ray comparison. Differential gene expression data is available at https://doi.org/10.5281/zenodo.13869958.

## Identification of candidate genes

To identify candidate *CoOSC*, *CoCYP* and *CoACT* genes, the pot marigold genome was searched using (i) *T. koksaghyz* taraxasterol synthase (GenBank ID: AXU93516) (ii) CYP716A111 (APG38190.1) and (iii) THAA3 (ASAT1; At3g51970.1) as queries in tBLASTn. Cut-off values were selected to include at least five expressed genes (present in the transcriptome datasets). The E value cut offs for gene identification were 1e-50 (TXSS), 4.52e-76 (*CYP*), and 1e-60 (*ACT*). To identify all OSCs in pot marigold, CoTXSS was used as query with the first exon (E value cut-off: 2.43e-98), last exon (E value cut-off:1.66e-45), and the exon containing the catalytic motif DCTAE (E value cut-off:8.49e-57). All retrieved sequences were translated investigated using InterPro (103.0), which identified OSCs as squalene cyclases (IPR018333), CYPs as cytochrome P450s (IPR001128) and ACTs as MBOAT wax synthases (IPR032805). The publicly available genomes of *Artemisia annua* (txid:35608), *Lactuca sativa* (txid:4236), *Cichorium endive* (txid:114280), *Taraxacum kok-saghyz* (txid:333970), *Helianthus anuus* (txid:4232), *Chrysanthemum seticuspe* (txid:1111766), *Cynara cardunculus* (txid:4265) and the *C. arvensis* transcriptome were similarly interrogated.

## Phylogenetic analysis

Protein sequences were aligned using MUSCLE V3.8.425[58] and sites with gaps were trimmed using ClipKIT with smartgap mode (v2.1.3)[59]. Maximum likelihood phylogenetic trees were inferred using IQ-TREE for all TXSS, CYP and ACT trees. For the OSC and ACT phylogenies, a JTT matrix-based model allowing for invariable sites plus discrete Gamma model and 1000 bootstraps was used (OSC and ACT model: JTT + F + I + G4)[60]. For construction of the CYP phylogeny, a LG model

with empirical amino acid frequencies plus a discrete Gamma model and 1000 bootstraps was used (CYP model: LG + F + I + G4). All models were selected using ModelFinder[61].

For Bayesian phylogenetic analysis, DNA sequences were aligned by codons, and all gaps were trimmed. BEAUTi2 was used to split the alignment into three partitions and generate an input file. The clock models and tree were linked among the three partitions. Analysis was performed using BEAST2[62] with a GTR model and an *Artemisia* fossil[63] and a *Cichorium intybus* type fossil[64] as calibrators[1]. A strict molecular clock with rate=1 was used. A Yule Model was used as a prior model and 10,000,000 rounds of MCMC were run with 10% of burn-in. The ESS of each parameter was verified using Tracer[65] and the maximum clade credibility tree was generated with TreeAnnotator. 95% length HPD was used to represent the branch length range.

## Construction of plant gene expression vectors

Candidate coding sequences were chemically synthesised (Twist Bioscience, South San Francisco, CA, USA) introducing synonymous mutations to remove recognition sites for BpiI, BsaI, BsmBI and SapI in accordance with the phytobrick standard[66]. Coding sequences were assembled into a binary backbone (pICH47732; Addgene #48000) together with a CaMV35s promoter and the omega sequence from tobacco mosaic virus (TMV) (pICH51277; Addgene #50268), and CaMV 35s terminator (pICH41414; Addgene #50337). Constructs were assembled using a one-step digestion-ligation reaction as previously described[67]. Sequence-verified constructs were transformed into *Agrobacterium tumefaciens* GV3101 by electroporation. Site directed mutagenesis of coding sequences was performed on expression constructs using the previously reported golden mutagenesis method[68]. A list of plasmids is provided in Supplementary Data 11; DNA and full sequences have been deposited in the Addgene repository (227509-227578).

## Transient expression in *Nicotiana benthamiana*

Single colonies of *A. tumefaciens* GV3101 containing candidate and control genes as well as the suppressor of gene silencing, P19 from Tomato Bushy Stunt Virus (TBSV), (pEPQD1CB0104; #177038)[69], and a feedback-insensitive form of HMG-CoA reductase, tHMGR (pEPQD1CB0817; #177039)[34] were used to inoculate liquid 20 ml LB with 50 μg/mL rifampicin, 20 μg/ml gentamicin, and 100 μg/ml carbenicillin. Overnight saturated cultures were centrifuged at $3400 \times g$ for 30 min at room temperature and cells were resuspended in infiltration medium (10 mM 2-(N-morpholino)ethanesulfonic acid (MES) pH 5.7, 10 mM MgCl2, 200 μM 3′,5′-Dimethoxy-4′-hydroxyacetophenone (acetosyringone)) and incubated at room temperature for 2-3 h at 100 rpm. Resuspended cultures were diluted to 0.8 OD$_{600}$ and then mixed in equal volumes to a final volume of 2 ml. Cultures were infiltrated into a nick in the abaxial surface of leaves of young (4 true leaves; 29–34 days old) *N. benthamiana* plants using a 1 ml needleless syringe. Infiltrated plants were grown at 22 °C in a MLR-352-PE plant growth chamber (Panasonic Healthcare Co, Oizumi-Machi, Japan) with a 16 h light, 8 h dark cycle for five days.

## Positive selection analysis

TXSS peptide sequences were aligned using MAFFT[70] and their codons were mapped to the alignment using pal2nal and all gaps in the alignment were trimmed. A maximum likelihood tree was constructed as above. The non-basal Cichorieae TXSSs were selected as the foreground. A null model (fix_omega = 1, omega = 1) and alternative model (fix_omega = 0, omega = 1) of branch-site model A were fitted to the phylogenetic tree and codon alignment using codeml function of PAML package[71]. A likelihood ratio test was used to evaluate positive selection[72] and Bayes Empirical Bayes was used to infer residues under positive selection[73].

## Structure modelling

Structural models of CoOSC3, CoTXSS, TkTXSS, CoCYP716A392 and CoCYP716A393 were constructed using AlphaFold2[24]. The best ranked models were used for docking. For TXSS, a taraxasteryl cation structural model was made based on 3D structure of taraxasterol (PubChem ID: 115250) and optimised in Gaussian with HF/6-31* basis set. TXSS structural models were aligned to the crystal structure of human lanosterol synthase with lanosterol in its active site (PDB:1W6K) using the align function of PyMOL (Schrödinger, Inc., New York, NY, USA) and the taraxasteryl cation was manually docked to the active site based on the location of lanosterol in 1W6K. Energy minimisation was performed with AMBER22[74]. Protein structural models of CoCYP716A392 and CoCYP716A393 were aligned to the crystal structure of CYP90B1A in a complex with cholesterol (PDB:6A15)[26] and Ψ-taraxasterol (PubChem ID: 115250) was manually docked to the active site based on the location of cholesterol in 6A15. Energy minimisation was performed with YASARA[75]. All structures were visualised in PyMOL.

## Quantitative reverse transcription PCR

Reverse transcription was carried out using the M-MLV cDNA synthesis system (Sigma: M1302) in a 12 μl reaction with 300 ng/μl of total RNA, 1 μl of dNTP mix (10 mM) and 1 μl of oligo dT$_{12-18}$ (0.5 μg/ml). Reactions were heated to 65 °C then cooled to 4 °C before addition of 4 μl of first-strand buffer (5X), 2 μl of DTT (0.1 M) and 1 μl of RNaseOUT™ (40 units/μl). Samples were incubated at 37 °C for 2 min before addition of water (control) or 200 U M-MLV reverse transcriptase and incubated at 37 °C for 50 min followed by inactivation at 70 °C for 15 min. cDNA was diluted 1:10 in TE Buffer (5 mM Tris-HCl + 0.5 mM EDTA, pH 8.0) before use in amplification reactions. *SAND* (SAND family protein), *PROTEIN PHOSPHATASE 2 A* and *PHOSPHOGLYCERATE KINASE* were previously identified as appropriate reference genes in *Chrysanthemum morifolium*[76,77]. Orthologues of these genes were identified in the pot marigold genome and their variance was assessed in the transcriptome data. From this analysis, *CoSAND* was selected as the most appropriate reference gene. Primers (Integrated DNA Technologies, Coralville, IA, US) were tested for efficiency (95 % - 105 %) and a melt curve with a single peak (Supplementary Data 12) using a QuantStudio™ 6 Pro Real-Time PCR system (Applied Biosystems, Waltham, MA, USA). Amplification was performed in 10 μl reactions with 0.2 μM each primer, 6 ng cDNA and 1 μl SYBR Green Jumpstart™ Taq ReadyMix™. Reactions were cycled at 94 °C for 2 min, followed by 40 cycles of 94 °C for 15 s and 58 °C for 60 s, followed by a melt curve. For each reaction, two technical replicates (qPCR reactions) of at least three biological replicates were performed. Control reactions (no reverse transcriptase; no template controls) had Cq values of >35. Relative expression analysis was carried out using the ΔΔCt method with timepoint S1 time point used for normalisation.

## Statistics and reproducibility

Analyses including metabolic profiling of Asteraceae species, transcriptomics, gene expression by qPCR, cell viability and anti-inflammatory assays were performed on a minimum of three biological replicates. Due to the variability of agroinfiltration, these experiments were performed on six biological replicates. No data were excluded. The Investigators were not blinded to allocation during experiments and outcome assessment. No statistical method was used to predetermine sample size.

## Reporting summary

Further information on research design is available in the Nature Portfolio Reporting Summary linked to this article.

# Data availability

The assembled genome of *C. officinalis* generated in this study is available under accession number GCA_964273985.1 [https://www.

ncbi.nlm.nih.gov/datasets/genome/GCA_964273985.1/]. The functional annotation of the *C. officinalis* genome and transcriptome assemblies of *C. officinalis* and *C. arvensis* are available at https://doi.org/10.5281/zenodo.13869958. Reads are available under accession numbers: PRJEB80524; [https://www.ebi.ac.uk/ena/browser/view/PRJEB80524] (*C. officinalis*) and PRJEB80545; [https://www.ebi.ac.uk/ena/browser/view/PRJEB80545] (*C. arvensis*). Plasmids are available at Addgene (227509-227578). Differential gene expression data, GC-MS data, LC-MS data, and NMR data are available at https://doi.org/10.5281/zenodo.13869958.

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

## Acknowledgements

We gratefully acknowledge the support of the Biotechnology and Biological Sciences Research Council (BBSRC), part of UK Research and Innovation (UKRI) for funding via grants BB/W014173/1 and the Earlham Institute strategic programme grant, Decoding Biodiversity (BBX011089/1) and its constituent work package (BBS/E/ER/230002B). Part of this work was delivered by Transformative Genomics, a BBSRC-funded National Bioscience Research Infrastructure (BBS/E/ER/23NB0006). DG is supported by a scholarship from the John Innes Foundation; HS is supported by a BBSRC NRP-DTP scholarship (BB/T008717/1 Project No. 2578291). The plasmid pLO-AstHMGR, containing the coding sequence of truncated HMGR from *A. strigosa* as well as β-amyrin, α-amyrin, lupeol, oleanolic acid and betulinic acid standards, were a kind gift from Anne Osbourn, John Innes Centre. *C. arvensis* seeds (Serial No. 32133) were gratefully retrieved from the Millennium Seed Bank. *T. kok-saghyz* seeds (Serial No. #W635156) were obtained from the National Plant Germplasm System Germplasm Resources Information Network (GRIN). We additionally thank Earlham Institute Transformative Genomics for library preparation and sequencing; Sergey Nepogodiev at the John Innes Centre NMR facility for

assistance with NMR; the John Innes Centre horticultural services team for help with plant husbandry; the John Innes Centre metabolomics facility for help with LC-MS and GC-MS and Lionel Hill for assistance with LC-MS analysis; Andrew Hemmings at the University of East Anglia for invaluable guidance with AlphaFold2 and for help and discussions on structural modelling and docking; Wilfried Haerty, Dave Wright and Will Nash at the Earlham Institute for advice and assistance with positive selection analysis; James Reed at the John Innes Centre for advice on triterpene detection; Ilia Leitch at the Royal Botanic Gardens at Kew for discussions on genome size and ploidy in the Asteraceae. We are grateful for the use of the ADA High Performance Computing facility at University of East Anglia. The funders had no role in the study design, data collection and analysis, decision to publish, or preparation of the manuscript.

## Author contributions

M.S., M.O'.C., and N.J.P. conceptualised the study. M.S. and H.S. performed metabolic profiling. D.G. and M.O'.C. performed all bioassays with human cell lines. M.S. extracted genomic DNA and RNA. G.G.K., G.L., C.S., and D.S. assembled the transcriptome and genome, and performed differential expression analysis. M.S., D.G., H.S., and C.T. identified candidate genes, performed phylogenetic analysis, cloned and performed function characterisation of candidate genes. H.S. and D.G. performed structural modelling and functional analysis of TXSS and CYP mutations with supervision from M.S. H.S. performed analyses for positive selection and Bayesian molecular clock analysis. M.S., D.G. and C.T. performed qRT-PCR. M.S., C.T., M.O'.C. and N.J.P. were responsible for supervision and project management. N.J.P. was responsible for fundraising. M.S., D.G., H.S., C.T., G.G.K., D.S., M.O'.C. and N.J.P. drafted the figures and text. All authors contributed to revisions and editing.

## Competing interests

The authors declare no competing interests.
