## [Transparent Peer Review file · Nature Communications]

Biosynthesis and Bioactivity of Anti-Inflammatory Triterpenoids in *Calendula officinalis*

Corresponding Author: Dr Nicola Patron

Version 0:

Reviewer comments:

Reviewer #1

(Remarks to the Author)

The manuscript "Biosynthesis and Bioactivity of Anti-Inflammatory Triterpenoids in *Calendula officinalis* (pot marigold)" explores the anti-inflammatory properties and biosynthetic pathway of triterpenoids in *Calendula officinalis*. The researchers first identified C16-hydroxylated triterpenoids, especially faradiol fatty acid esters, as key contributors to the anti-inflammatory activity of pot marigold floral extracts. They found that these compounds inhibited the release of pro-inflammatory cytokines TNF- α and IL-6 in LPS-activated human monocytic cells. Mechanistically, faradiol regulated IL-6 production by reducing the phosphorylation of STAT3. Next, they uncovered the genetic basis of triterpenoid production. Candidate genes encoding oxidosqualene synthases (OSCs), cytochromes P450s (CYPs), and acyltransferases (ACTs) were identified through genome and transcriptome sequencing, phylogenetic analysis, and functional characterization. Finally, they reconstructed the complete biosynthetic pathway in *Nicotiana benthamiana*, providing a platform for the production of anti-inflammatory components. This study highlights the potential of integrated bioactivity and biosynthesis studies to unlock the therapeutic potential of medicinal plants. I like the two mutant experiments very much, they explained the biosynthetic differ of and amyrin/taraxasterol and Ψ -taraxasterol/taraxasterol.

There are some concerns about this manuscript as followings:

1. The article seems more like two distinct pieces of work, especially the first part "The Anti-inflammatory Activity of C:16 Hydroxylated Triterpenoids," which is a standalone project and the research is relatively preliminary, requiring a more systematic explanation.
2. For the OSCs finding, I think maybe use ortholog analysis, or at least blast + pfam will get more comprehensive sequences.
3. Line 204, do synteny regions conserved in other Asteraceae plants?
4. Most importantly, although the authors have done a substantial amount of work, it feels like they have not fully explored its biological significance.

Reviewer #2

(Remarks to the Author)

The manuscript of Golubova and coworkers describes the elucidation of both the anti-inflammatory properties of triterpenoids from *Calendula officinalis* and their biosynthetic modalities. This plant, commonly known as pot marigold, is a medicinal plant from the Asteraceae family whose extracts are currently exploited for skin care. Interestingly, pot marigold accumulates different triterpenoids including faradiol palmitate as the most abundant one, with many of them that may contribute to the anti-inflammatory effects of the plant extract. However, the precise role of each triterpenoid in these properties as well as their mechanisms of action remained poorly known. In the first part of their work, the author established by measuring pro-inflammatory cytokine release that faradiol, faradiol palmitate and other fatty acid esters are key contributors to the anti-inflammatory activity of pot marigold extracts through their action of the interleukin-6 secretion. They also established that this effect was mediated by reducing phosphorylation of the transcription factors STAT3. Next, by sequencing the genome of *C. officinalis* and performing associated transcriptomics analysis, they initiated the elucidation of the biosynthetic pathway of these triterpenoids. By predicting gene candidate using homology searches and performing functional assays in through transient expression in *Nicotiana benthamiana*, they identified an oxidosqualene cyclase (OSC, here named CoTXSS, a couple of P450 (CoCYP716A392 and CoCYP716A393) and a couple of acyltransferases (CoACT1 and CoACT2) that ensure the synthesis of faradiol palmitate from 2,3-oxidosqualene. Through structural model prediction,

they also identified residues important for their activities as revealed by mutational studies and propose hypotheses regarding evolution of these enzymes. Lastly, they have reconstituted the whole pathway through transient expression of the whole set of gene in *N. benthamiana*. All experiments have been well conducted and the results are convincing. The manuscript is well written and elegantly highlights the main discoveries of this work. Comments/concerns are appended below:

- Introduction: Including a figure describing the biosynthetic pathway and the structure of the mentioned molecules would ease understanding and facilitate reading
- For ψ -taraxasterol synthases, they authors mentioned that they identified 27 CoOSC candidates through homology searches but only retrieved the transcripts of 16 of them in their transcriptome. Why such a difference? wrong gene annotation? No corresponding gene expression? Do the seven CoOSCs found in the clade of proteins involved in plant sterol biosynthesis strictly belong to the 16 candidates found in the transcriptome. Can the authors clarify this point?
- The authors proposed that CoTXSS evolved from CoMAS by duplication and neofunctionalization, which is certainly right. What about their respective position in the genome? Any additional evidence based on this?
- Based on their functional assays (sup Figure 20), the CarTXSS, HaTXSS, CcTXSS, TdTXSS seems more efficient than CoTXSS (considering this type of assay is partially quantitative). Why did the author use CoTXSS for full pathway reconstitution in bent? Can we expect an increase in faradiol palmitate using the aforementioned enzymes?
- Same question regarding the difference between the P450 candidates retrieved from genome and transcriptome? Clarification would ease understanding.
- Regarding the characterization of the P450 mutant, the A284G mutation seems to have the opposite effect on the faradiol production of CoCYP716A392 and CoCYP716A393. Can the authors comment on this difference?
- Figure 6: including the quantification of 2,3-oxidosqualene, ψ -taraxasterol, faradiol and faradiol palmitate at the same six developmental stages would be interesting.
- Via pathway reconstitution in benth, the authors produced around 1.3 μg of faraiol per mg of dry weight which is 3.5-fold more than its amount in pot marigold floral extract. Using benth, production cost is undoubtedly higher than that of a faradiol purified from natural resources. Can the authors estimate the titer they need to reach in benth to get a system economically viable compared to pot marigold? Do they envisage producing this compound in other host systems such as microbial cell factories?

Reviewer #3

(Remarks to the Author)

The manuscript titled "Biosynthesis and Bioactivity of Anti-Inflammatory Triterpenoids in *Calendula officinalis* (pot marigold)" presents a comprehensive investigation into the biosynthesis and bioactivity of triterpenoids in *Calendula officinalis*. The study provides valuable insights into the anti-inflammatory properties of these compounds, particularly focusing on C16-hydroxylated triterpenoids. The authors have successfully elucidated the biosynthetic pathway and reconstructed it in *Nicotiana benthamiana*, offering a platform for the production of these bioactive compounds. This work presents a significant discovery in the field, but certain methodological and presentational aspects require clarification and strengthening to meet the journal's high standards. Major revisions are necessary to address the following concerns.

1, The use of tobacco leaf injection for enzyme activity assays is appropriate for qualitative functional screening. However, quantitative comparisons of enzyme activity (e.g., via point mutations) require standardized protein quantities. To strengthen the conclusions, perform *in vitro* assays using purified proteins expressed in *E. coli* or yeast, ensuring equal protein amounts are used.

2, The description of bacterial co-inoculation in the tobacco experiment is unclear. Specifically: Did the authors mix bacterial cultures after adjusting each strain to $\text{OD}_{600} = 0.8$, or did they first mix the strains and then adjust the OD? If multiple strains are mixed post-OD adjustment, the effective concentration of each strain would be diluted to $0.8/n$, potentially reducing synthesis efficiency. To ensure consistency, should each strain should be adjusted to $\text{OD}_{600} = 0.8 \times n$ before mixing? Clarify this step in the Methods section and provide a rationale for the chosen approach.

3, While the supplemental materials include the compound identities corresponding to the chromatographic peaks in Figure 1, the main figure lacks numerical labels for key compounds. To improve readability, please number the major peaks in Figure 1 and list the corresponding compound names in the figure legend. This adjustment will enhance accessibility for readers and align with best practices in data visualization

4. The authors mention the successful reconstruction of the biosynthetic pathway in *Nicotiana benthamiana* but do not provide specific data on the yield of the bioactive compounds. Including detailed data on the yield and potential scalability of this production platform would strengthen the practical implications of their findings.

5, Some sections could benefit from clearer transitions and more concise language to enhance the overall flow of the paper.

Version 1:

Reviewer comments:

Reviewer #1

(Remarks to the Author)

The author has provided responses and discussions on the issues. I respect the author's efforts during the exploration process, but I still have some comments for reference.

1. The author appears to address two issues in this paper: one is "The anti-inflammatory activity of C:16 hydroxylated triterpenoids," and the other is its biosynthesis. In reality, these two issues are not inherently linked. Forcing them together only makes the paper appear lengthy and inadequately explained. For example, the author emphasizes "we note that none of the reviewers has disputed this finding or the methods." I have already pointed out that the current work is preliminary and insufficient to support the subsequent logic. From the methods used by the author, they only employed one cell line without conducting extensive testing, lacking animal experiments or organoid experiments. Additionally, the author did not analyze the changes of different compounds after entering the bloodstream, which further makes this speculation untenable. The author cites the work on artemisinin and lists some literatures. I suspect the author may lack an understanding of the discovery process of artemisinin. Although its mechanism was published recently, within one to two years after its isolation, artemisinin completed cytological experiments, animal experiments, and clinical trials, with multiple process improvements during this period. This example precisely illustrates that confirming the physiological function of a compound is a systematic effort. The author has cited many good literatures, but they should also refer to Professor Tu Youyou's Nobel Lecture and related work introductions (Tu Y. Artemisinin-A Gift from Traditional Chinese Medicine to the World (Nobel Lecture). *Angew Chem Int Ed Engl.* 2016 Aug 22;55(35):10210-26. doi: 10.1002/anie.201601967.).

2. Following up on the previous issue, due to structural problems, many descriptions of biosynthetic discovery in this paper are insufficient or even confusing. For instance, in the genome section, there seem to be unclear basic concepts. The author did not introduce the ploidy of the species but directly listed the busco duplication rate (Supplementary Tab5) and did not explain whether the splicing results were redundant. This made it very difficult for me to track the relevant genes in this paper. For example, the author mentions "diploidization" (line 168), and I only understood it might refer to "tetraploidization" because "homeologue" appeared later. If it is tetraploidization, this would cause problems in the author's comparison of gene numbers. For example, the author argues that the number of OSCs is 没问题 (no problem) because it is consistent with many diploid genomes, which is illogical. Conversely, if the author does not mean tetraploidization, it indicates problems with their genome splicing, such as an excessively high duplication rate, and the current gene count remains problematic. Of course, it may also be caused by WGD (whole-genome duplication) or WGT (whole-genome triplication), but I have not seen a clear analysis of this in the paper. When pointing out the genome, the author should not only provide the txid but also cite the genome version and related papers. Early-published genomes and recently published ones differ significantly in sequencing, splicing, and analysis methods, and they are sometimes incomparable.

3. Regarding the issue of collinearity, in the previous version, I mainly wanted to ask about the many gene duplications discovered by the author and understand whether these gene duplications (whether tandem or scattered repeats) occurred before or after species differentiation, so collinearity needs to be examined. The author actually provided a good example, such as CoTXSS and CoMAS. I am also curious about other gene pairs, but the author only mentioned the situation within the published genome. It is also unclear whether these gene pairs were generated by WGD or WGT. Additionally, in line 188, the author concludes that genes are "non-functional pseudogenes" because no expression was detected, despite using "likely," this statement is still too arbitrary.

4. Regarding the response to question 4, the author presented their opinions and listed many literatures. Conversely, many opposing literatures could also be cited, such as "Boccia et al doi: 10.1126/science.ado3409." From another perspective, if the author placed the cytological experiments after synthetic biology, it might be more convincing and better illustrate the biological significance. If such a shift were made, some protein experiments might need to be added, such as SPR or pull-down, to clarify the target.

Reviewer #2

(Remarks to the Author)

In the revised version of their manuscript, the authors have addressed some of the concerns raised by my reviewing. This is an interesting work.

Reviewer #3

(Remarks to the Author)

My comments are properly addressed.

Point-by-point response to reviewer comments.

Reviewer #1

The manuscript "Biosynthesis and Bioactivity of Anti-Inflammatory Triterpenoids in *Calendula officinalis* (pot marigold)" explores the anti-inflammatory properties and biosynthetic pathway of triterpenoids in *Calendula officinalis*. The researchers first identified C16-hydroxylated triterpenoids, especially faradiol fatty acid esters, as key contributors to the anti-inflammatory activity of pot marigold floral extracts. They found that these compounds inhibited the release of pro-inflammatory cytokines TNF- α and IL-6 in LPS-activated human monocytic cells. Mechanistically, faradiol regulated IL-6 production by reducing the phosphorylation of STAT3. Next, they uncovered the genetic basis of triterpenoid production. Candidate genes encoding oxidosqualene synthases (OSCs), cytochromes P450s (CYPs), and acyltransferases (ACTs) were identified through genome and transcriptome sequencing, phylogenetic analysis, and functional characterization. Finally, they reconstructed the complete biosynthetic pathway in *Nicotiana benthamiana*, providing a platform for the production of anti-inflammatory components. This study highlights the potential of integrated bioactivity and biosynthesis studies to unlock the therapeutic potential of medicinal plants. I like the two mutant experiments very much, they explained the biosynthetic differ of and amyirin/taraxasterol and Ψ -taraxasterol/taraxasterol.

We thank the reviewer for their time and for noting their acknowledgement of integrating bioactivity and biosynthesis studies to unlock the therapeutic potential of medicinal plants and the mutation experiments.

There are some concerns about this manuscript as followings:

1. The article seems more like two distinct pieces of work, especially the first part "The Anti-inflammatory Activity of C:16 Hydroxylated Triterpenoids," which is a standalone project and the research is relatively preliminary, requiring a more systematic explanation.

We first note that this comment is at odds with the reviewer's previous comment on 'the value of integrating bioactivity and biosynthesis studies'.

To clarify our aim, the first section of the manuscript "The Anti-inflammatory Activity of C:16 Hydroxylated Triterpenoids" provides the motivation for subsequent gene discovery and enzyme characterisation. While we agree that very few previous papers have included both the bioactivity of a molecule and the gene discovery of the pathway, we disagree that these are standalone studies: without first confirming that these molecules are responsible for the anti-inflammatory activity, there would be little justification for determining the

genetic basis of these molecules or reconstituting their biosynthetic pathways.

Regarding the preliminary nature of this first section, we reject the notion that the anti-inflammatory assays are preliminary. These assays are detailed and conclusively demonstrate the anti-inflammatory activity of C:16 hydroxylated esters. We note that none of the reviewers has disputed this finding or the methods. We agree that this discovery provides a foundation for future systematic investigations of the molecular mechanism of the bioactivity. However, such studies will take many years and will comprise an independent line of investigation. We note that a systematic understanding of the mechanism of action of many drugs is often not gained until long after their bioactivity is conclusively reported. For example, insights into the mechanism by which artemisinin kills the malaria parasite were not published until 2018 (Bridgford et al <https://doi.org/10.1038/s41467-018-06221-1>), which was 4 decades after the bioactivity of the purified compound was first shown, and 12 years after biosynthesis of the molecule had been reconstructed in yeast (Ro *et al.* <https://doi.org/10.1038/nature04640>).

In response to this comment, and in line with the suggestion of Reviewer 3 to make the transitions between sections smoother, we have made revisions to provide a better flow between the results sections.

2. For the OSCs finding, I think maybe **use ortholog analysis, or at least blast + pfam** will get more comprehensive sequences.

We thank the reviewer for this comment but perhaps they overlooked that methods section, “Identification of candidate genes”, which describes the use of candidate ortholog identification including BLAST etc.. Subsequently, we described the use of phylogenomic inference analyses and, in the results text, we also reported the protein families, including the MBOAT family for ACTs. To ensure this is clear, we have revised to include a further sentence in the methods “All retrieved sequences were investigated using InterPro (103.0) which identified OSCs as squalene cyclases (IPR018333), CYPs as cytochrome P450s (IPR001128) and ACTs as MBOAT wax synthases (IPR032805).”

We are confident that we have comprehensively identified all OSCs using these methods: The number of OSC-encoding genes that we identified in the tetraploid genome of pot marigold (28) is comparable to that found in other plant species — For example, 25 OSCs were identified in the tetraploid *Brassica napus* (Liu et al., 2019). Similarly, 13 OSCs were found in the diploid genome of *Avena strigosa* (Liang et al., 2022); 14 in the diploid genome of *Arabidopsis thaliana* (Liu et al., 2019); 13 in the diploid genome of *Carica*

papaya (Liu et al., 2019) , and 12 in the diploid genome of *Oryza sativa* (Ma et al., 2024). Thus, the number of OSC genes is consistent with expectations.

3. Line 204, **do synteny regions conserved in other Asteraceae plants?**

We think that the reviewer is asking if the region in which the TXSS gene is located shows synteny with other Asteraceae genomes. We have investigated this, and found that the syntenic organisation of this genomic region is conserved in other Asteraceae. This data is presented within a new Supplementary figure (Supplementary Figure 22) and a description added to the manuscript, later in this section of text where we present the other data regarding evolutionary and synteny.

4. Most importantly, although the authors have done a substantial amount of work, it feels like they have not fully explored its biological significance.

The non-specific nature of this comment makes it very difficult to respond to. Biological significance could refer to more than one thing within this manuscript and the reviewer has not given any indication of what they refer to or any direction of what should be improved.

If the reviewer is referring to the function of the molecule as an anti-inflammatory, as noted above, we believe that these experiments are conclusive in that we have shown that specific molecules are bioactive, which provides the motivation to uncover their biosynthetic pathway as a route to enabling heterologous biosynthesis. As per our response above, systematic investigations of the specific mechanisms by which bioactives work, often take years/decades and are beyond the scope of this study.

If the reviewer is referring to the biological function in the plant, then this is beyond the scope of the study, as has proven the case for many other molecules, will be incredibly difficult to confirm: The function within the host plant of the vast majority of plant specialised metabolites, including exceptionally well-studied molecules such as artemisinin, are still the subject of speculation and that there are numerous reviews on the difficulty of investigating the biological significance of plant specialised metabolites - e.g. Gargounas et al, 2021 (<https://doi.org/10.1111/nph.17470>); Huand and Dudareva, 2023 (<https://doi.org/10.1016/j.cub.2023.01.057>); Kozmakz et al 2020 (<https://doi.org/10.1016/j.pbi.2020.02.005>). While there are numerous reasons that this work is challenging, one of the most important is that, for the vast majority of plants, we lack knowledge of the specific pests, pathogens

and stressors that are/were present in the specific geographic regions in which they evolved.

Reviewer #2 (Remarks to the Author)

The manuscript of Golubova and coworkers describes the elucidation of both the anti-inflammatory properties of triterpenoids from *Calendula officinalis* and their biosynthetic modalities. This plant, commonly known as pot marigold, is a medicinal plant from the Asteraceae family whose extracts are currently exploited for skin care. Interestingly, pot marigold accumulates different triterpenoids including faradiol palmitate as the most abundant one, with many of them that may contribute to the anti-inflammatory effects of the plant extract. However, the precise role of each triterpenoid in these properties as well as their mechanisms of action remained poorly known. In the first part of their work, the author established by measuring pro-inflammatory cytokine release that faradiol, faradiol palmitate and other fatty acid esters are key contributors to the anti-inflammatory activity of pot marigold extracts through their action of the interleukin-6 secretion. They also established that this effect was mediated by reducing phosphorylation of the transcription factors STAT3. Next, by sequencing the genome of *C. officinalis* and performing associated transcriptomics analysis, they initiated the elucidation of the biosynthetic pathway of these triterpenoids. By predicting gene candidate using homology searches and performing functional assays in through transient expression in *Nicotiana benthamiana*, they identified an oxidosqualene cyclase (OSC, here named CoTXSS, a couple of P450 (CoCYP716A392 and CoCYP716A393) and a couple of acyltransferases (CoACT1 and CoACT2) that ensure the synthesis of faradiol palmitate from 2,3-oxidosqualene. Through structural model prediction, they also identified residues important for their activities as revealed by mutational studies and propose hypotheses regarding evolution of these enzymes. Lastly, they have reconstituted the whole pathway through transient expression of the whole set of gene in *N. benthamiana*. All experiments have been well conducted and the results are convincing. The manuscript is well written and elegantly highlights the main discoveries of this work. Comments/concerns are appended below:

We thank the reviewer for their work and note in particular that they found the experiments have been well conducted and the results to be convincing.

1. Introduction: Including a figure describing the biosynthetic pathway and the structure of the mentioned molecules would ease understanding and facilitate reading

Thank you for the suggestion. We have added a new figure (Figure 1) showing the structure of the molecules studied in the manuscript to the introduction. It would be premature to include details of enzymes at this point in the manuscript as their discover is not described until the results.

- For ψ -taraxasterol synthases, they authors mentioned that they identified 27 CoOSC candidates through homology searches but only retrieved the transcripts of 16 of them in their transcriptome. Why such a difference? wrong gene annotation? No corresponding gene expression? Do the seven CoOSCs found in the clade of proteins involved in plant sterol biosynthesis strictly belong to the 16 candidates found in the transcriptome. Can the authors clarify this point?

We did not expect to find transcripts of all OSCs encoded in the genome within our transcriptomes. This is because: (i) we did not conduct transcriptome analysis of all tissues at all stages of development. It is likely that some OSCs are only expressed in, for example, roots, and others might only expressed at specific stages of development, (ii) as the species is an ancient tetraploid, we expected some homeologues to have accumulated stop codons due to diploidization and therefore no longer expressed - indeed, this is what we observed and described in the results; (iii) it is widely observed that many specialised metabolic pathways are only expressed in response to specific biotic pathogens or abiotic stress conditions.

OSCs involved in sterol biosynthesis are included these 16, as discussed in the results.

To clarify, we have revised the text to include a sentence about the OSCs for which expression was not detected: "The expression patterns of other CoOSCs for which transcripts were not present in our leaf and floral transcriptomes may be limited to other tissues (e.g., roots) or in response to specific stimuli."

- The authors proposed that CoTXSS evolved from CoMAS by duplication and neofunctionalization, which is certainly right. What about their respective position in the genome? Any additional evidence based on this?

We thank the reviewer for the question. The existing Supplementary Figure 18 already showed that CoTXSS and CoMAS are in different genomic locations. The phylogenetic analysis in Figure 2 (now Figure 3) indicates that there are 5 closely related MAS genes in *C. officinalis*, including the gene we characterised and named CoMAS. Indeed, CoTXSS is co-located with one of these - OSC2 (Supplementary Figure 18). We did not clone and characterise as we did not find evidence of expression. Nevertheless, this does provide additional evidence that CoTXSS evolved by duplication. Further, we find that the syntenic organisation of this genomic region is conserved in other Asteraceae, which is now shown in a new Supplementary Figure (Supplementary Figure 22), to address the question raised by Reviewer 1. We

have added text to the end of the section entitled “TXSSs evolved in the Asteraceae from a multifunctional amyrin synthase and have been maintained in all major lineages” to describe the colocalization of TXSS and MAS and the conservation of synteny on other Asteraceae.

- Based on their functional assays (sup Figure 20), the CarTXSS, HaTXSS, CcTXSS, TdTXSS seems more efficient than CoTXSS (considering this type of assay is partially quantitative). Why did the author use CoTXSS for full pathway reconstitution in bent? Can we expect an increase in faradiol palmitate using the aforementioned enzymes?

First, we note that the functional analysis of CarTXSS, HaTXSS, CcTXSS, TdTXSS are not quantitative and so we cannot conclude that they are more efficient. We did test alternative TXSSs for several reasons: First, this paper focussed on identification on a bioactive compound and the elucidation of its pathway. In future studies, multiple metabolic engineering strategies will need to be compared. Second, from the results we obtain, it can be seen that more significant limitations in the yields of faradiol/faradiol palmitate result from (i) the relatively low activity of the CYPs, which were unable to completely convert the available taraxasterol/psi-taraxasterol scaffolds into the diol and, (ii) the loss of diols by, most likely, derivatization by endogenous enzymes.

- Same question regarding the difference between the P450 candidates retrieved from genome and transcriptome? Clarification would ease understanding.

The same applies to CYPs as to TXSS – we do not expect all gene to be expressed in the leaves/flowers. To clarify, we have added a sentence: “The expression of the other genes may be limited to other tissues or specific conditions.”

- Regarding the characterization of the P450 mutant, the A284G mutation seems to have the opposite effect on the faradiol production of CoCYP716A392 and CoCYP716A393. Can the authors comment on this difference?

In Supplementary Figure 30 (now 31) the A284G mutation of both CoCYP716A392 and CoCYP716A393 had a similar effect - a reduction in activity: As described in the text, product abundance as a measure substrate specificity was unreliable due to loss of some product to derivatisation by endogenous enzymes and thus we quantified the depletion of the substrates, psi-taraxasterol and beta-amyrin (Supplementary Figure 31 B). In both CoCYP716A392 and CoCYP716A393, there is less substrate depletion (more psi-taraxasterol and beta-amyrin remaining) than for the WT enzymes. We

believe that this is well described in the results text and figure legend but have made some edits to ensure clarity.

- Figure 6: including the quantification of 2,3-oxidosqualene, ψ -taraxasterol, faradiol and fardiol palmitate at the same six developmental stages would be interesting.

We agree with the reviewer and thank them for their interest. We have added a new supplementary figure (Supplementary Figure 37) quantifying the metabolites through development and included sentences to describe these results in the text.

- Via pathway reconstitution in benth, the authors produced around 1.3 μg of faraiol per mg of dry weight which is 3.5-fold more than its amount in pot marigold floral extract. Using benth, production cost is undoubtedly higher than that of a faradiol purified from natural resources. Can the authors estimate the titer they need to reach in benth to get a system economically viable compared to pot marigold? Do they envisage producing this compound in other host systems such as microbial cell factories?

Estimating a viable yield is well beyond what is currently possible. *N. benthamiana* has yet to be demonstrated for the commercial scale production of metabolites and, to the best of our knowledge, no life-cycle analyses on metabolite production in this species are publicly available. Therefore, there are no figures on which to estimate viable yields. In addition, as the bioactive compound had not previously been conclusively identified, there is no established/scaled process for the extraction from pot marigold with which to compare the cost of production in *N. benthamiana*. Further, the yields we have reported for pot marigold are from specific floral tissues. There are no target products in the green tissues (the majority of harvested biomass) as deconstruction of flowers to maximise yield would also be laborious and expensive. Finally, this manuscript does not explore the numerous and extensive avenues that could be taken to maximise yields in *N. benthamiana*; this would be an entirely separate (and very lengthy) investigation.

For similar reasons, we cannot speculate on which production chassis will be the most suitable. Production of triterpenoids in bacterial hosts is known to be particularly challenging due to the association of key enzymes (OSCs and CYPs) with the endoplasmic reticulum (see also our response to review 3). While the production of some complex triterpenoids has recently been produced in yeast (e.g. Guo et al 2020 <https://pubs.acs.org/doi/10.1021/acssynbio.0c00124>), it is extremely difficult to compare yields from liquid culture and plants. Indeed, we recently published a review article (Golubova et al 2024; <https://doi.org/10.1016/j.pbi.2024.102611>

for review) in which we discussed the immense challenges of trying to equivocate yields from microbes (often given in g/L) to yields from plants (per unit dry or fresh weight). This is especially pertinent when the monetary and environmental costs of feedstocks (sugars extracted from cultivated plants) compared to water and light. Without life-cycle analyses, yield comparisons of yields are misleading and such comparisons are beyond the scope of this manuscript which is not focussed on metabolic engineering yield optimisation.

We have not included such discussions in the manuscript as our manuscript is focussed on the identification of an anti-inflammatory bioactives and the enzymes in its biosynthetic pathway. Importantly, we note that other recent and equivalent manuscripts that describe the elucidation of metabolic pathways in *N. benthamiana*, including those with known market values have not estimated viable yields for heterologous bioproduction. For example: podophyllotoxin (Lau and Sattely, 2025, <https://doi.org/10.1126/science.aac7202>), cocaine (Wang et al 2022; <https://doi.org/10.1021/jacs.2c09091>) and Momilactone B (de la Pena and Sattley 2021; <https://doi.org/10.1038/s41589-020-00669-3>). However, to ensure that our manuscript is not misleading, we have added text to make it clear that future studies will need to be conducted to identify the best method of production of these molecules and to optimise their yields.

Reviewer #3 (Remarks to the Author)

The manuscript titled "Biosynthesis and Bioactivity of Anti-Inflammatory Triterpenoids in *Calendula officinalis* (pot marigold)" presents a comprehensive investigation into the biosynthesis and bioactivity of triterpenoids in *Calendula officinalis*. The study provides valuable insights into the anti-inflammatory properties of these compounds, particularly focusing on C16-hydroxylated triterpenoids. The authors have successfully elucidated the biosynthetic pathway and reconstructed it in *Nicotiana benthamiana*, offering a platform for the production of these bioactive compounds. This work presents a significant discovery in the field, but certain methodological and presentational aspects require clarification and strengthening to meet the journal's high standards. Major revisions are necessary to address the following concerns.

We thank the reviewer for noting that this work presents a significant discovery in the field.

1. The use of tobacco leaf injection for enzyme activity assays is appropriate for qualitative functional screening. However, quantitative comparisons of enzyme activity (e.g., via point mutations) require standardized protein quantities. To strengthen the conclusions, perform in vitro assays using

purified proteins expressed in *E. coli* or yeast, ensuring equal protein amounts are used.

It is not technically feasible to express these particular classes of enzymes in bacterial hosts and neither is it possible to purify them from yeasts. This is very widely described in the literature. For this reason, the methods we have used the same methods as those reported in other studies that have investigated other pathways that include enzymes from these larger families:

Plant OSCs are monotopic membrane proteins in which the substrate entrance platform is embedded in the outer ER membrane enabling easy access to its non-polar substrate, 2,3-oxidosqualene, produced by ER-attached squalene epoxidase. We note that there is only a single crystal structure of an OSC (human lanosterol synthase, PDB ID: 1W6K), which is indicative of the difficulty of obtaining purified protein. Negative results, unfortunately, are seldom published. However, the purification of this enzyme class has been extensively investigated in other labs, as evidenced by unsuccessful attempts to express plant OSCs in *E. coli* and to purify them from yeast described in several PhD theses, e.g Dokarry 2010 (<https://ueaeprints.uea.ac.uk/id/eprint/20511/1/2010DokarryMPhD.pdf>) (Chapters 4 and 5) and Pfalzgraf 2022 (<https://ueaeprints.uea.ac.uk/id/eprint/91656/1/HP%20220127%20PhD%20Thesis%20-%20redacted%20%28available%20for%20public%29.pdf>) (Chapter 5).

In contrast, to investigate the determinants of product specificity in OSCs that produce hopanoids, Liang et al., 2022; (<https://www.pnas.org/doi/full/10.1073/pnas.2118709119>) used the same approach as us and performed reciprocal mutagenesis on hopanoid synthases in *N. benthamiana*, quantifying metabolite production with an internal standard. Similarly, to investigate the importance of VFM/VFN motif in beta-amyrin and cycloartenol synthases, Chen et al., 2023 (<https://www.sciencedirect.com/science/article/pii/S2090123222000765#s0070>) performed mutagenesis on eight plant OSCs, which were expressed in both yeast and *N. benthamiana* and metabolite production quantified with internal standards. They observed similar results between yeast and *N. benthamiana*, however, Srisawat et al., 2019 (<https://nph.onlinelibrary.wiley.com/doi/10.1111/nph.16013>), who similarly characterised alpha-amyrin synthases in both yeast and *N. benthamiana* reported that *N. benthamiana* is a more suitable host than yeast. Other examples of plant OSC mutagenesis and characterisation are summarised in Chen et al., 2021 (<https://pubs.rsc.org/en/content/articlelanding/2021/np/d1np00015b>).

Plant CYP proteins are also membrane-bound enzymes, mostly localised in the endoplasmic reticulum (ER) membrane, plastid envelope, or membranes of other organelles (Zhou et al 2021; <https://www.sciencedirect.com/science/article/pii/S1369526621000054>). Their activity often depends on membrane localisation and interaction with redox partners, which are absent or poorly functional in prokaryotic systems. Consistent with this, to the best of our knowledge, although many hundreds of plant CYPs have been identified, there are only seven crystal structures, highlighting the difficulty of obtaining active, soluble forms of these enzymes.

Site-directed mutagenesis with subsequent product quantification in *Nicotiana benthamiana* is also therefore often used for characterising product or substrate specificity of plant CYPs. For example, Geisler et al 2013 (<https://doi.org/10.1073/pnas.1309157110>) used this method to characterise AsCYP51H10 from *Avena sativa* (oat) and Gnanasekaran, et al 2015 (<https://jbioleng.biomedcentral.com/articles/10.1186/s13036-015-0022-z>) used this method to characterise CYP720B4, quantifying products using the internal standard eicosane.

2. The description of bacterial co-inoculation in the tobacco experiment is unclear. Specifically: Did the authors mix bacterial cultures after adjusting each strain to OD600 = 0.8, or did they first mix the strains and then adjust the OD? If multiple strains are mixed post-OD adjustment, the effective concentration of each strain would be diluted to 0.8/n, potentially reducing synthesis efficiency. To ensure consistency, should each strain should be adjusted to OD600 = 0.8 × n before mixing? Clarify this step in the Methods section and provide a rationale for the chosen approach.

Multiple strains were adjusted to an OD600 of 0.8 then subsequently mixed in equal volumes to ensure equal delivery of different constructs. This could not be achieved by mixing the strains and then adjusting OD. Previous studies have shown that, providing the OD600 of each strain is >0.2, there will be near simultaneous delivery to all cells (Carlson et al., 2023; <https://pubs.acs.org/doi/10.1021/acssynbio.3c00148>). In contrast, it is important that the final mixture infiltrated into leaves does not exceed OD600 0.8 as this can cause necrosis. Therefore, we do not adjust strains to OD600 = 0.8 × n. To make the method clearer, we have revised the text to read: “Resuspended cultures were diluted to 0.8 OD₆₀₀, then equal volumes of these cultures were combined to obtain a final volume of 2 ml”.

We also note, that as per our response to reviewer 2, this manuscript does not explore the numerous and extensive avenues that could be taken to

maximise yields; this would be an entirely separate (and very lengthy) investigation.

3, While the supplemental materials include the compound identities corresponding to the chromatographic peaks in Figure 1, the main figure lacks numerical labels for key compounds. To improve readability, please number the major peaks in Figure 1 and list the corresponding compound names in the figure legend. This adjustment will enhance accessibility for readers and align with best practices in data visualization

We have added numbers of the peaks and a list of corresponding compound names to the figure legend. We note that this is now Figure 2.

4. The authors mention the successful reconstruction of the biosynthetic pathway in *Nicotiana benthamiana* but do not provide specific data on the yield of the bioactive compounds. Including detailed data on the yield and potential scalability of this production platform would strengthen the practical implications of their findings.

The original version of the manuscript did include yields, as commented on by Reviewer 2. We have now revised the text to the following information, which also integrates the yield data from the new Supplementary Figure 37: “It also enabled us to reconstruct both biosynthesis of faradiol palmitate and that of the highly bioactive intermediate, faradiol. For the latter, we obtained yields of up to 2.34 µg/mg dw (Figure 7), 9.4-fold higher than those found in extracts of mature pot marigold flowers (0.25 µg/mg dw; Supplementary Figure 37) and 3.8-fold higher than those found in developing buds (0.61 µg/mg dw; Supplementary Figure 37) (Supplementary table 1). We note that these are the initial yields of pathway reconstruction. Future metabolic engineering efforts are required to optimise yields and to identify the best methods and chassis in which to produce these compounds.”

Similar to our response to reviewer 2, as per other studies that utilise the advantages of *N. benthamiana* for the elucidation of plant biosynthetic pathways, we are not claiming that this is the best or final production platform or that these are the maximal yields. We have revised the text to ensure that the text is clear that the identification of the bioactive compounds and the knowledge of its pathway is a starting point for future metabolic engineering to enable access. Such engineering efforts include multiple and extensive engineering of both the pathway and of host metabolism - in plant systems these are complex endeavours taking several years - and are an entirely separate undertaking.

5, Some sections could benefit from clearer transitions and more concise language to enhance the overall flow of the paper.

As per the response to reviewer 1, we have revised the transitions between manuscript sections to enhance flow.

Point-by-point response to reviewer comments.

Reviewer #1

The author has provided responses and discussions on the issues. I respect the author's efforts during the exploration process, but I still have some comments for reference.

1. The author appears to address two issues in this paper: one is "The anti-inflammatory activity of C:16 hydroxylated triterpenoids," and the other is its biosynthesis. In reality, these two issues are not inherently linked. Forcing them together only makes the paper appear lengthy and inadequately explained. For example, the author emphasizes "we note that none of the reviewers has disputed this finding or the methods." I have already pointed out that the current work is preliminary and insufficient to support the subsequent logic. From the methods used by the author, they only employed one cell line without conducting extensive testing, lacking animal experiments or organoid experiments. Additionally, the author did not analyze the changes of different compounds after entering the bloodstream, which further makes this speculation untenable. The author cites the work on artemisinin and lists some literatures. I suspect the author may lack an understanding of the discovery process of artemisinin. Although its mechanism was published recently, within one to two years after its isolation, artemisinin completed cytological experiments, animal experiments, and clinical trials, with multiple process improvements during this period. This example precisely illustrates that confirming the physiological function of a compound is a systematic effort. The author has cited many good literatures, but they should also refer to Professor Tu Youyou's Nobel Lecture and related work introductions (Tu Y. Artemisinin- A Gift from Traditional Chinese Medicine to the World (Nobel Lecture). *Angew Chem Int Ed Engl.* 2016 Aug 22;55(35):10210-26. doi: 10.1002/anie.201601967.).

Response. We respectfully disagree that the current work is preliminary and insufficient or that stories have been forced together. We note that the text cites to previous preliminary work performed in other cell lines and in animal models. Our findings are consistent with these previous studies but go further and are more conclusive are far from preliminary. We agree that many more experiments can be performed to further investigate the mechanisms and developed it as a drug. However, experiments with organoids, cytological experiments, animal experiments, and clinical trials are numerous years of work and are completely beyond the scope of this manuscript.

2. Following up on the previous issue, due to structural problems, many descriptions of biosynthetic discovery in this paper are insufficient or even confusing. For instance, in the genome section, there seem to be unclear basic concepts. The author did not introduce the ploidy of the species but directly listed the BUSCO duplication rate (Supplementary Tab5) and did not explain whether the splicing results were redundant. This made it very difficult for me to track the relevant genes in this paper. For example, the author mentions "diploidization" (line 168), and I only understood it might refer to "tetraploidization" because "homeologue" appeared later. If it is tetraploidization, this would cause problems in the author's comparison of gene numbers. For example, the author argues that the number of OSCs is 没问题 (no problem) because it is consistent with many diploid genomes, which is illogical. Conversely, if the author does not mean tetraploidization, it indicates problems with their genome splicing, such as an excessively high duplication rate, and the current gene count remains problematic. Of course, it may also be caused by WGD (whole-genome duplication) or WGT (whole-genome triplication), but I have not seen a clear analysis of this in the paper.

When pointing out the genome, the author should not only provide the txid but also cite the genome version and related papers. Early-published genomes and recently published ones differ significantly in sequencing, splicing, and analysis methods, and they are sometimes incomparable.

Response:

To make the ploidy of the species explicitly clear, **we have added a further sentence (line 168) and citation to previous literature stating that *C. officinalis* is an allotetraploid.** We note that the K-mer and BUSCO analyses support this and so everything is as expected. We also note that in our previous response we explicitly stated that the 28 OSC-encoding genes that we identified in the **tetraploid** genome of pot marigold is comparable to that found in other **tetraploid** plant species e.g. 25 OSCs in the tetraploid *Brassica napus* (Liu et al., 2019). We also stated that diploid genomes have roughly half this number. Thus, the comparison is perfectly logical and there is no problem with the gene number. We do not know what genome splicing is - we can only assume that the reviewer is referring to alternative splicing, we are unclear why this is relevant as Kmer or BUSCOs analyses are performed on genomes not the transcriptomes?

Our use of diploidisation (the evolutionary process where a polyploid genome is reduced back to a diploid state) is accurate. The K-mer analysis, the BUSCO count, and also by the fact that some homologues have accumulated mutations are consistent with diploidisation of a tetraploid. We have stated that "one of each gene pair was not expressed and comparative analysis of the intron-exon structure, sequence and conserved catalytic motifs indicated these non-expressed OSCs were likely to be non-functional pseudogenes". This is highly consistent with equivalent observations in a very large number of polyploid plants and the text already cites a review (Li, Z. et al. *Annu Rev Plant Biol* **72**, 387–410 (2021)).

We are afraid that we don't know what the reviewer is referring to in their comment "did not explain whether the splicing results were redundant". We do not show splicing results.

Supplementary Data 5 shows the statistics for the genome assembly. We are unsure what these would be redundant with?

The reviewer has overlooked that the pot marigold genome was sequenced and assembled for this manuscript – a link to the raw data and the assembly are provided as part of the accompanying data. We do not compare old and new genome assemblies and as we didn't use any other *C. officinalis* genome versions, there are no related papers or previous genome versions to cite. The genomes of other species were not analysed except in Supplementary Data 22. We don't refer to these genomes in the main text. However, **we have added the genome versions used for this analysis to Supplementary Data 22.**

3. Regarding the issue of collinearity, in the previous version, I mainly wanted to ask about the many gene duplications discovered by the author and understand whether these gene duplications (whether tandem or scattered repeats) occurred before or after species differentiation, so collinearity needs to be examined. The author actually provided a good example, such as CoTXSS and CoMAS. I am also curious about other gene pairs, but the author only mentioned the situation within the published genome. It is also unclear whether these gene pairs were generated by WGD or WGT. Additionally, in line 188, the author concludes that genes are "non-functional pseudogenes" because no expression was detected, despite using "likely," this statement is still too arbitrary.

Response: This has already been addressed. The two gene copies appear to be on homoeologous chromosomes. This is consistent with the species being a tetraploid thus it is self-evident that the gene pairs were created in the allotetraploidisation event. As the diploid ancestors of this species have not been conclusively identified (and, of course, might be extinct!), we cannot look at or compare their genomes to *C. officinalis*. For the other gene pairs, the manuscript already includes data to show that gene pairs are also on homoeologous chromosomes (Supp figs 28 and 36).

We note that we do not classify genes as "non-functional pseudogenes" because no expression was detected. The reviewer has overlooked that the text explicitly refers to mutations that render many of those genes unlikely to be functional. The specific mutations found in each gene which led to this conclusion are already detailed in column 7 of Supplementary Table 6. As per our response to the previous question, these are classic signatures of diploidisation and expected in a tetraploid.

4. Regarding the response to question 4, the author presented their opinions and listed many literatures. Conversely, many opposing literatures could also be cited, such as "Boccia et al doi: 10.1126/science.ado3409." From another perspective, if the author placed the cytological experiments after synthetic biology, it might be more convincing and better illustrate the biological significance. If such a shift were made, some protein experiments might need to be added, such as SPR or pull-down, to clarify the target.

Response: The paper cited by the reviewer (Boccia et al doi: 10.1126/science.ado3409) does not report the discovery of the biological function of the molecule. As stated in the first few sentences of that paper, “SGAs are toxic antinutrients (e.g., α -tomatine in tomato, α -solanine and α -chaconine in potato) with well-characterized roles in plant defense”. Thus, the paper focusses on identification of a key aspect of their biosynthesis pathway. The authors were then able to demonstrate the importance of the key enzyme by implementing a new version of a feeding assay with knock-out plants.

In our case, the biological function of triterpene fatty acid esters in plants is unknown. As per our previous response: The function within the host plant of the vast majority of plant specialised metabolites, including exceptionally well-studied molecules, are still the subject of speculation and that there are numerous reviews on the difficulty of investigating the biological significance of plant specialised metabolites - e.g. Gargounas et al, 2021 (<https://doi.org/10.1111/nph.17470>); Huand and Dudareva, 2023 (<https://doi.org/10.1016/j.cub.2023.01.057>); Kozmakz et al 2020 (<https://doi.org/10.1016/j.pbi.2020.02.005>).

Reviewer #2 (Remarks to the Author):

In the revised version of their manuscript, the authors have addressed some of the concerns raised by my reviewing.

This is an interesting work.

Reviewer #3 (Remarks to the Author):

My comments are properly addressed.